# Cytomegalovirus late transcription factor target sequence diversity orchestrates viral early to late transcription

Ming Li[1,2&], Qiaolin Hu[1,2&], Geoffrey Collins[3], Mrutyunjaya Parida[3], Christopher B. Ball[3], David H. Price[3], Jeffery L. Meier[1,2,4]*

1 Iowa City Veterans Affairs Health Care System, Iowa City, Iowa, United States of America, 2 Department of Internal Medicine University of Iowa, Iowa City, Iowa, United States of America, 3 Department of Biochemistry, University of Iowa, Iowa City, Iowa, United States of America, 4 Department of Epidemiology, University of Iowa, Iowa City, Iowa, United States of America

& These authors contributed equally to this work.
* Jeffery-meier@uiowa.edu

**Data Availability Statement:** The Github links to view aligned reads for PRO-Seq and ChIP-Seq datasets are available at: https://github.com/meierjl/Towne-UL87LTF https://github.com/meierjl/

## Abstract

Beta- and gammaherpesviruses late transcription factors (LTFs) target viral promoters containing a TATT sequence to drive transcription after viral DNA replication has begun. Human cytomegalovirus (HCMV), a betaherpesvirus, uses the UL87 LTF to bind both TATT and host RNA polymerase II (Pol II), whereas the UL79 LTF has been suggested to drive productive elongation. Here we apply integrated functional genomics (dTag system, PRO-Seq, ChIP-Seq, and promoter function assays) to uncover the contribution of diversity in LTF target sequences in determining degree and scope to which LTFs drive viral transcription. We characterize the DNA sequence patterns in LTF-responsive and -unresponsive promoter populations, determine where and when Pol II initiates transcription, identify sites of LTF binding genome-wide, and quantify change in nascent transcripts from individual promoters in relation to core promoter sequences, LTF loss, stage of infection, and viral DNA replication. We find that HCMV UL79 and UL87 LTFs function concordantly to initiate transcription from over half of all active viral promoters in late infection, while not appreciably affecting host transcription. Both LTFs act on and bind to viral early-late and late kinetic-class promoters. Over one-third of these core promoters lack the TATT and instead have a TATAT, TGTT, or YRYT. The TATT and non-TATT motifs are part of a sequence block with a sequence code that correlates with promoter transcription level. LTF occupancy of a TATATA palindrome shared by back-to-back promoters is linked to bidirectional transcription. We conclude that diversity in LTF target sequences shapes the LTF-transformative program that drives the viral early-to-late transcription switch.

## Author summary

Herpesviruses have a group of genes earmarked for expression late in the infection. Beta- and gammaherpesviruses utilize a six-member set of viral late transcription factors to

TB40E-UL79LTF Source data is provided for all figures and analyses presented in this manuscript. PRO-Seq and ChIP-Seq datasets are available at the Gene Expression Omnibus (accession: GSE168165).

**Funding:** This research was supported by the Department of Veterans Affairs Merit award I01 BX004434 to J.L.M., National Institute of Allergy and Infectious Diseases R21-AI130453 to J.L.M. and D.H.P., National Institute of General Medical Science R35-GM126908 to D.H.P, National Institute of Allergy and Infectious Diseases T32-AI007533 (C.B.B). The funders had no role in study design, data collection, and analysis, decision to publish, or preparation of the manuscript.

**Competing interests:** The authors have declared that no competing interests exist.

selectively activate these genes by binding to a DNA sequence signature in gene promoters. We made an unexpected discovery that a wider range of differences in sequence signatures configures the late gene expression program for human cytomegalovirus, a beta-herpesvirus of global public health importance. Diversity in signature patterns expands promoter targets and probably pre-sets amount of individual promoter output. A unique palindromic sequence signature is linked to the activation of back-to-back promoters at multiple locations in the viral genome. We deduce that diversity in late transcription factor targets functionally orchestrates the rollout of a productive late-stage infection. This may be a generalizable feature adopted by beta- and gammaherpesvirus subfamilies.

## Introduction

The family *Herpesviridae* are large double-stranded DNA viruses. Divergent evolution has produced *Alpha-*, *Beta-*, *and Gammaherpesvirinae* subfamilies. Mammals, birds, and reptiles are natural hosts for alphaherpesviruses, while mammals are the only known hosts for beta- and gammaherpesviruses [1,2]. Humans are host to alphaherpesviruses, herpes simplex viruses 1 and 2; betaherpesviruses, human cytomegalovirus (HCMV) and human herpesviruses 6 and 7; and gammaherpesviruses, Epstein-Barr virus (EBV) and Kaposi sarcoma-associated herpesvirus (KSHV). All herpesviruses produce lytic and latent infections. Lytic infection produces new infectious virions via the coordinate regulation of three sequential programs of viral gene expression that make immediate-early (IE), early, and late gene products. The IE, early, and late gene expression programs differ in the mechanisms that regulate them. Viral IE gene expression is brought about by host transcription factors that bind cis-regulatory sequences in IE gene promoters, as well as the actions of virion-associated proteins. Viral IE gene products activate expression of viral early genes with the assistance of host transcription factors that bind cis-regulatory sequences in early gene promoters. A set of viral IE and early proteins carry out viral DNA synthesis that consequently enables viral late gene promoters to become active. These viral late gene promoters do not have the cis-regulatory sequences to which host transcription factors bind.

Beta-and gammaherpesviruses apply the same general strategy to activate their late gene promoters after the onset of viral DNA replication. This entails use of a TATT sequence in place of a TATA-box as a binding site for a six-member set of viral late transcription factors (LTF) that bring about transcription. One of the 6 LTF members specializes in attaching the LTF complex to the TATT sequence [3,4] while also associating with the unphosphorylated carboxy terminal domain of host RNA polymerase II (Pol II) [5]. The association of the LTF assembly with host general transcription factors, i.e., TFIIB and TFIIH, presents the functional equivalent of a host preinitiation complex [4]. The HCMV LTF that serves as a TATT-binding protein is encoded by the UL87 gene [4]. In HCMV-infected cells, the UL87 LTF co-immunoprecipitates with UL79 and UL95 LTFs, as well as host Pol II [6]. The HCMV UL79 LTF has been implicated as a positive regulator of Pol II productive elongation, and not involved in Pol II recruitment to the promoter [6]. The KSHV LTF homologs corresponding to HCMV UL79 and UL91 LTFs physically interact to activate late promoters, and mutating two conserved amino acid residues in the HCMV UL79 LTF disrupts the UL79-UL91 LTF interaction [7]. The HCMV UL95 LTF may be a scaffolding factor at the hub in the LTF complex, as suggested from protein-protein interaction studies of murine CMV m95 LTF [8] and KSHV ORF34 LTF homologs [7]. Phosphorylation of the EBV LTF homolog for HCMV UL95 LTF stabilizes the LTF complex [9], whereas phosphorylation of the EBV LTF homolog for HCMV UL92 LTF

prevents this LTF's degradation [10]. Five members of the 6-member set of HCMV LTFs (UL79, UL87, UL91, UL92, and UL95 LTFs) have been individually studied and shown to be essential for RNA expression from a select set of viral late promoters [11–14].

For both beta- and gammaherpesviruses, a change in mature mRNA level resulting from the absence of an LTF has been the standard measurement used to identify an LTF-activated viral gene [4,11–20]. Possible hidden effects on processing or stability of transcripts complicates the interpretation of results produced by this type of measurement. To circumvent the confounding caused by overlapping polygenic transcripts that are common in these viruses, a recent study characterized LTF-dependent EBV promoters by measuring levels of 5'-ends of mature capped mRNAs at promoters in wild-type (WT), BDLF4 LTF-null, and DNA replication-defective EBV genomes [21]. This study determined that the TATTWAA (W = A or T) sequence is enriched in LTF-dependent late promoters, as well as LTF-dependent early-late kinetic class promoters having transcription start sites (TSS) active at low levels in early infection and at higher levels after viral DNA replication [21]. KSHV LTF-dependent viral promoters have also been recently characterized genome-wide by comparing WT virus with a virus having a functionally inactive ORF24 LTF, which normally binds to the TATT and is the homolog of the HCMV UL87 LTF [22]. Measurements of KSHV late gene RNA expression by RNA-Seq applied to total RNA revealed that LTFs regulate viral late and early-late promoters. While the TATTWAA was found to be enriched in both kinetic classes of promoters, not all LTF-dependent promoters had the TATTWAA and not all TATTWAA sequences were involved in LTF-dependent RNA expression. Surprisingly, absence of the ORF24 LTF impaired viral KSHV DNA replication that was associated with a ORF24 binding element in the viral origin of DNA replication and required for late gene promoter activation [22]. To distinguish between direct and indirect effects of LTF loss on KSHV promoters, ChIP-Seq was used to detect LTF occupancy of LTF-responsive promoters, and the MEME motif analysis tool was applied to determine the LTF binding sequences that were enriched in this set of promoters. This uncovered a 5-bp motif immediately downstream of the TATTWAA. Base-substitution mutagenesis of the 5-bp GGGAC sequence flanking the TATTAAA in the KSHV late K8.1 promoter decreased both promoter activity and LTF binding to the promoter, when assayed in a reporter plasmid transfected into KSHV-infected cells. This study, as well as other studies, have demonstrated that simply changing the TATT to TATA also decreases LTF-dependent promoter activity, which is linked to a decrease in LTF binding to the TATA [23]. Whether the KSHV 5-bp motif requirement for enhanced ORF 24 LTF binding applies to betaherpesviruses is unknown, given that beta- and gammaherpesvirus LTF homologs differ greatly in primary amino acid structure and cannot be mixed and matched between these herpesvirus subfamilies. The KSHV study uncovered an LTF-driven promoter that differed from the others in having a TATAT instead of a TATT motif [22]. Unknown is the extent to which LTF target sequence variation contributes to viral transcription regulation.

Here we leverage recent advances in technologies to delineate the previously unrecognized contribution of HCMV LTF target sequence diversity in expanding the pool of LTF targets and likely determining individual promoter output. We demonstrate that HCMV UL79 and UL87 LTFs jointly function in transcription initiation at viral early-late and late promoters in late infection. Surprisingly, over one-third of these core promoters lack the TATT. These noncanonical promoters usually have a TATAT, TGTT, or YRYT motif. We also show that the TATT and non-TATT motifs are part of a larger sequence block with a sequence code that is predictive of promoter strength. LTF occupancy of a TATATA palindrome is linked to bidirectional transcription. The results support a model whereby LTF target sequence diversity configures the regulatory program that orchestrates the viral early-to-late transcription switch.

## Results

### Rapid on-demand depletion of viral LTFs

To determine the effect of viral LTF loss on transcription across viral and host genomes, we used the dTag system [24,25] to individually degrade HCMV UL79 and UL87 LTFs in late infection in human foreskin fibroblasts (HFF). We had previously constructed HCMV Towne UL87$^H$F that has FKBP12$^{F36V}$ and hemagglutinin (HA) tags fused to the carboxy end of the UL87 LTF (Fig 1A) and performed a pilot study to optimize the dTag system [24]. Adding dTag1 degrader to HCMV UL87$^H$F-infected HFF for 6 h in late infection depleted the UL87 LTF [24] via the host proteasome after dTag1 joins the FKBP12$^{F36V}$ segment of the chimeric UL87 LTF to the host cereblon E3 ligase complex (Fig 1B). Because the dTag treatment had lowered the amount of mature RNA message produced by the virus's TATT-containing late UL82 promoter [26], we advanced our studies here by first performing validation testing to show that the untreated HCMV UL87$^H$F phenocopies the HCMV UL87$^H$ comparator lacking the larger FKBP12$^{F36V}$ tag, as well as phenocopies the wild type (WT) parent virus. These test findings confirmed that HCMV UL87$^H$F and UL87$^H$ are equivalent at high multiplicity of infection (MOI) in their rates of production of UL87 LTF, IE1, IE2, and late pp28 proteins and viral DNA genomes (S1A and S1B Fig). HCMV UL87$^H$F and WT parent virus genomes also

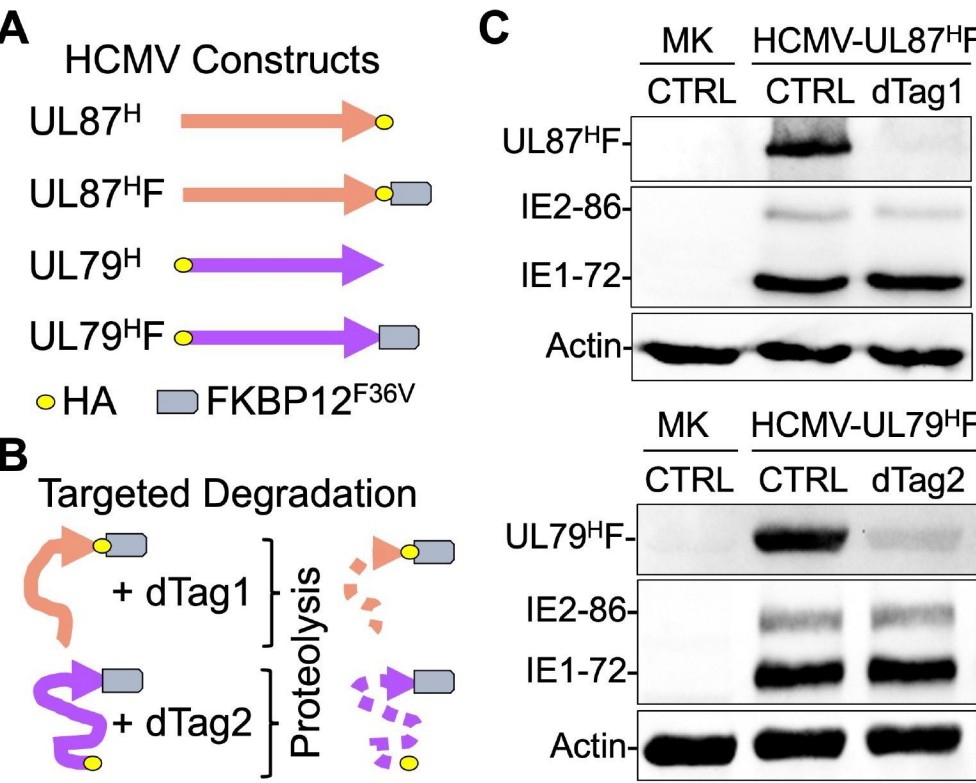

**Fig 1. Rapid depletion of UL79 and UL87 LTFs in late infection.** (**A**) The carboxy and amino ends of UL87 and UL79 LTFs (orange and violet arrows) of HCMV Towne and TB40/E strains, respectively, were tagged with the HA epitope. Fusing the FKBP12$^{F36V}$ tag to the carboxy ends of these LTFs created HCMV UL87$^H$F and UL79$^H$F constructs, whereas the comparator HCMV UL87$^H$ and UL79$^H$ constructs lack FKBP12$^{F36V}$. (**B**) dTag1 and dTag2 degraders were added for 6 h at the end of HCMV UL87$^H$F and UL79$^H$F infections (MOI of 3), respectively, to deplete FKBP12$^{F36V}$-tagged LTFs via proteolysis. (**C**) Western blot reveals that dTag treatments result in 94% and 87% depletion of UL87$^H$F and UL79$^H$F, respectively, at 96 and 72 h pi. Viral IE1-72 and IE2-86 protein levels did not change.

replicated to comparable levels (**S1C Fig**) and HCMV UL87^HF DNA replication was unaffected by the 6-h dTag1 treatment (**S1D Fig**). Because the laboratory HCMV Towne strain is attenuated and missing several genes, we selected the clinical-like HCMV TB40/E strain to study the outcome of UL79 LTF depletion. HCMV TB40/E UL79^H and HCMV UL79^HF were therefore constructed, in which an HA tag was placed at the amino end of UL79 LTF in both viruses and FKBP12^F36V was placed at the carboxy end of the UL79 LTF in HCMV UL79^HF (**Figs 1A, S2A, and S2B**). The FKBP12^F36V tag did not affect expression of UL79 LTF, IE1, IE2, and late pp28 proteins (**S2C Fig**) or alter viral DNA replication when compared to WT parent virus (**S2D Fig**), consistent with a previous study of a different tag on UL79 LTF [11]. During the course of optimizing conditions, we determined that dTag2 was more effective than dTag1 in degrading the UL79^HF in late infection. dTag2 is also a bifunctional molecule but differs from dTag1 in utilizing the Von Hippel-Lindau E3 ubiquitin ligase complex to degrade FKBP12^F36V-tagged proteins [27]. Having optimized the dTag system for disposing of UL79 and UL87 LTFs, separate sets of HFF were infected at high MOI with HCMV UL79^HF or HCMV UL87^HF, and each set of infections was divided for analyses to determine the degree of LTF degradation and the effects of LTF depletion on viral and host transcription. As shown in **Fig 1C**, the 6-h window of dTag1 and dTag2 treatments decreases the amounts of tagged UL79 and UL87 LTFs by 87% and 94%, respectively, in late infection. This narrow window of time in which the LTF depletion is achieved does not decrease amount of viral IE1 and IE2-86 proteins (**Fig 1C**) or late proteins IE2-40, IE2-60, and pp28 (**S1E** and **S2E Figs**).

## LTF depletion extensively affects viral transcription

We applied PRO-Seq to measure changes in transcription resulting from depleting LTFs in late infection. PRO-Seq has advantages over other methods by enabling the determination of precisely where host Pol II and its nascent transcript is located on host and viral genomes regardless of whether the transcript is capped, polyadenylated, or yields stable message [26,28,29]. It also distinguishes between effects on transcription initiation versus productive elongation. Halting Pol II productive elongation with flavopiridol (Flavo) treatment of infected cells mostly confines Pol II to the promoter-proximal pause region and eliminates confounding by polygenic transcription that is common across the viral genome. This PRO-Seq-Flavo method enables reliable measurements of amount of Pol II nascent RNA originating within a predefined 20-bp transcription start region (TSR). Each TSR arises from a maximum TSS (MAXTSS) and possibly smaller adjacent TSS. The number of MAXTSS reads is proportional to the number of reads at the TSR summit. A high number of TSR reads signifies robust TSS use. Independent analytical methods of PRO-Cap [29] and RNAse protection assay (RPA) [26] that measure 5'-ends of nascent and stable RNAs, respectively, corroborate the accuracy of PRO-Seq-Flavo measurements of TSS utilization. The outcome of late infection LTF depletion on viral transcription was determined for spike-in normalized PRO-Seq-Flavo reads aligned to the indicated HCMV genome that is viewable in the UCSC Genome browser using the following Github links to HCMV Towne (https://github.com/meierjl/Towne-UL87LTF) and TB40/E (https://github.com/meierjl/TB40E-UL79LTF) datasets. Each aligned read is derived from a unique (non-duplicative) Pol II nascent transcript. A genome-wide view of results from the same set of HCMV UL87^HF infections analyzed in **Fig 1C** reveals that the 6-h dTag1 treatment compared to vehicle control (CTRL) causes most viral TSRs to shrink in size or disappear (**Fig 2A**). We have previously reported that the same dTag1 dose applied to untagged HCMV WT for 90–96 hpi does not affect viral transcription [26]. While the depletion of UL87 LTF by dTag treatment has dramatic effects on viral transcription in late infection, not all viral promoter transcription is affected. As an example, the highly active HCMV RNA4.9 promoter,

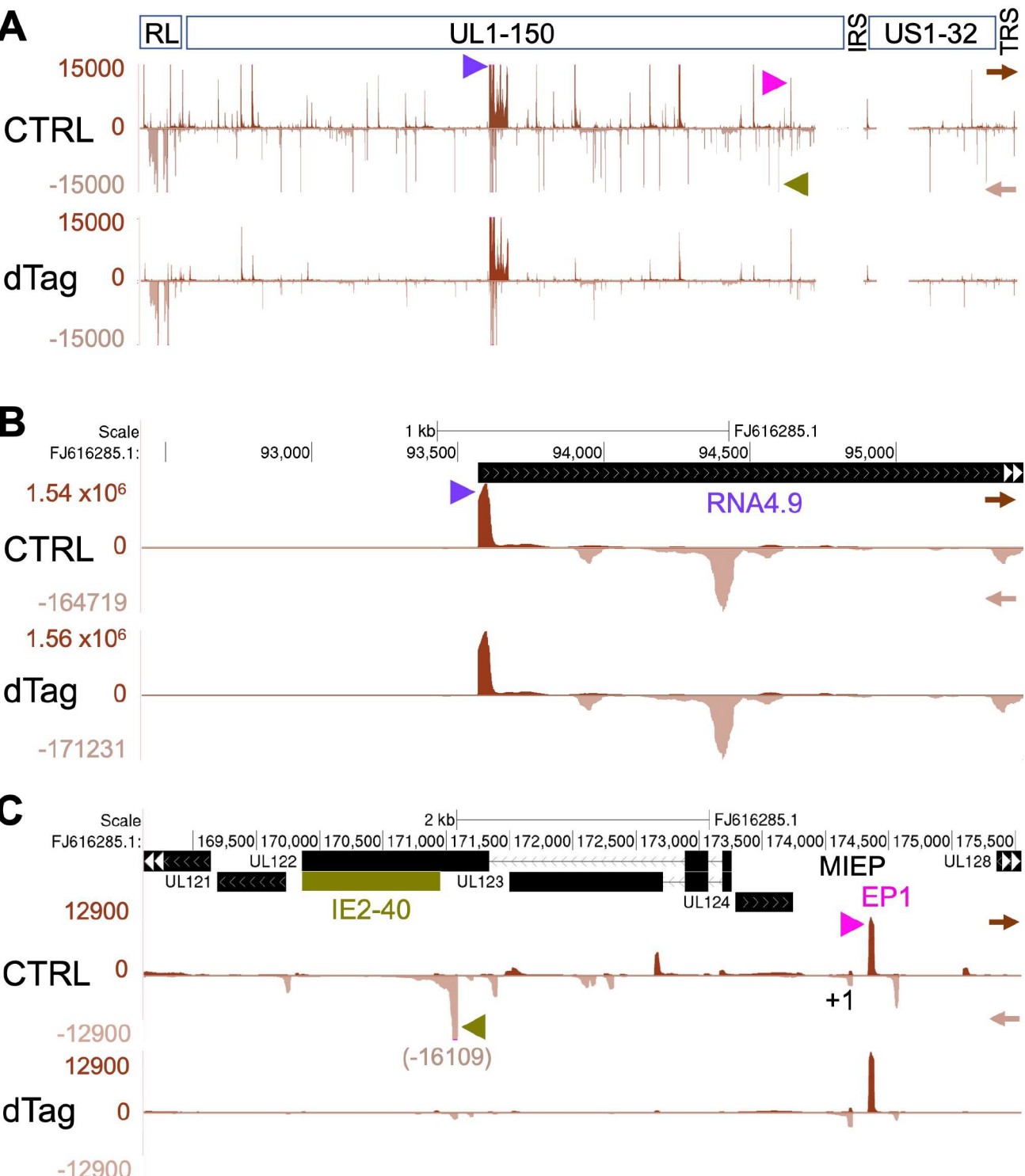

**Fig 2. LTF depletion effect on viral transcription initiation.** HFF infected with HCMV Towne UL87[H]F were treated with vehicle control (CTRL) or dTag1 (dTag) for 6h at 90–96 hpi and Pol II nascent transcripts were quantified by PRO-Seq-Flavo method. The spike-in normalized reads were aligned to the HCMV Towne FJ616285 genome. (**A**) UCSC Genome Browser view of the entire viral genome. Scale set at 15000 reads to view majority of viral TSRs. Track arrows point in direction of transcription. Arrow heads mark TSRs for RNA4.9 (violet), IE2-40 (olive green), and EP1 (pink). (**B**) Auto-scaled browser view focusing on RNA4.9 and antisense RNA4.9 TSRs. (**C**) Browser view of genome region containing MIEP (+1 TSS), IE2-40, and EP1 TSRs. Scale set at 12900 reads to better view majority of viral TSRs in this region.

with an astounding TSR strength of 1.5 million unique reads, is unaffected by UL87 LTF depletion (**Fig 2B**). The major immediate-early promoter (MIEP) TSR and the IE2-driven enhancer promoter (EP1) TSR are notable examples of other viral promoters that are unaffected by UL87 LTF depletion (**Fig 2C**). In contrast, the robust TSR for the neighboring late IE2-40 promoter is nearly abolished by the UL87 LTF depletion (**Fig 2C**).

Our initial study of using dTag1 treatment to deplete the UL87 LTF was carried out at 96 hpi to allow comparison of results to those produced by dTag1-induced depletion of viral IE2 proteins at the same late infection timepoint [26]. In the subsequent study of dTag2-induced UL79 LTF depletion, PRO-Seq-Flavo was applied at 72 hpi to better compare our results to those generated by other investigators using different methods that were applied at this late infection timepoint [6]. The genome browser views of the effects of UL79 LTF depletion at 72 hpi resulting from 6-h dTag2 treatment (66–72 hpi) show changes in viral transcription that are remarkably like the changes observed at 96 hpi because of depleting UL87 LTF for 6 h. Comparison of PRO-Seq results from parallel infections treated with vs. without Flavo revealed that productive elongation does not decrease downstream of viral promoters that remain active despite UL79$^{H}$F depletion, as viewable by the USCS Genome browser (https://github.com/meierjl/TB40E-UL79LTF) and exemplified in **S3 Fig**. We determined the extent to which UL79 and UL87 LTFs drive transcription from promoters across the viral genome. Viral TSRs producing >200 reads were selected for evaluation, so as to filter out extremely small TSRs and potential artifacts. Scatterplots display the change in TSR strength resulting from UL79 or UL87 LTF depletion in late infection vs. the viral TSR strength in the absence of the depletion (**Fig 3A and 3B**). A separation of LTF-responsive and unresponsive promoter populations is clear. The UL79 and UL87 LTFs drive nearly 60% of all viral TSRs in late infection. The TSRs in both promoter populations range in strength by 2 orders of magnitude, excluding the TSRs in the RNA4.9 region. TSRs for the LTF-responsive IE2-40 promoter and the LTF-unresponsive MIEP are selected as reference points for their respective populations. Of the 302 active TSRs conserved between Towne and TB40/E strains, ~64% of them require both UL79 and UL87 LTFs for activation during late infection (**Fig 3C** and **S1 Data**). The strength of 6 viral TSRs decreased >50% due to UL87 LTF depletion at 96 hpi but decreased 28–50% from UL79 LTF depletion at 72 hpi. Conversely, the strength of 3 viral TSRs decreased >50% due to UL79 LTF depletion and decreased 26–48% from UL87 LTF depletion. Based on findings of HCMV UL87$^{H}$F infections later carried out for 72 h using a different viral stock and HFF line, we determined the 1-day difference in time of late infection does not substantially affect the overall weight of results. Only 2.5% fewer viral TSRs decrease >50% in strength at 72 hpi compared to 96 hpi when depleting the UL87 LTF for the last 6 h of these infections (**S4A and S4B Fig**). Cross-comparing HCMV Towne UL87 LTF depletion results with those from the TB40/E UL79 LTF depletion revealed that the UL87 LTF depletions resulted in 1.5% and 2.6% more viral TSRs decreasing >50% at 72 and 96 hpi, respectively (**S4C Fig** and **S1 Data**), which is probably because UL87 LTF depletion is slightly more robust than that of UL79 LTF depletion (**Fig 1**). We determined the nucleotide positions of the MAXTSS for all active viral promoters in late infection by mapping the 5'-end of reads from the PRO-Seq-Flavo and prior PRO-Cap [29] datasets. This allowed us to confirm the LTF depletion had decreased the number of MAXTSS reads to a similar degree as that of the TSR reads and there was concordance in UL79 LTF vs. UL87 LTF depletion results by this measure (**S4D Fig**). LTF depletion had not decreased MAXTSS reads at the LTF-unresponsive promoter TSRs.

Knowing the exact nucleotide positions of MAXTSS for all LTF-responsive and -unresponsive TSRs enabled a comprehensive analysis of sequence motifs in the region -50 to -20 upstream of the MAXTSS of these promoters. The MEME discovery algorithm matched 279 of 279 UL87 LTF-responsive TSRs in generating an 8-nucleotide logo with a TATTW (W = A or

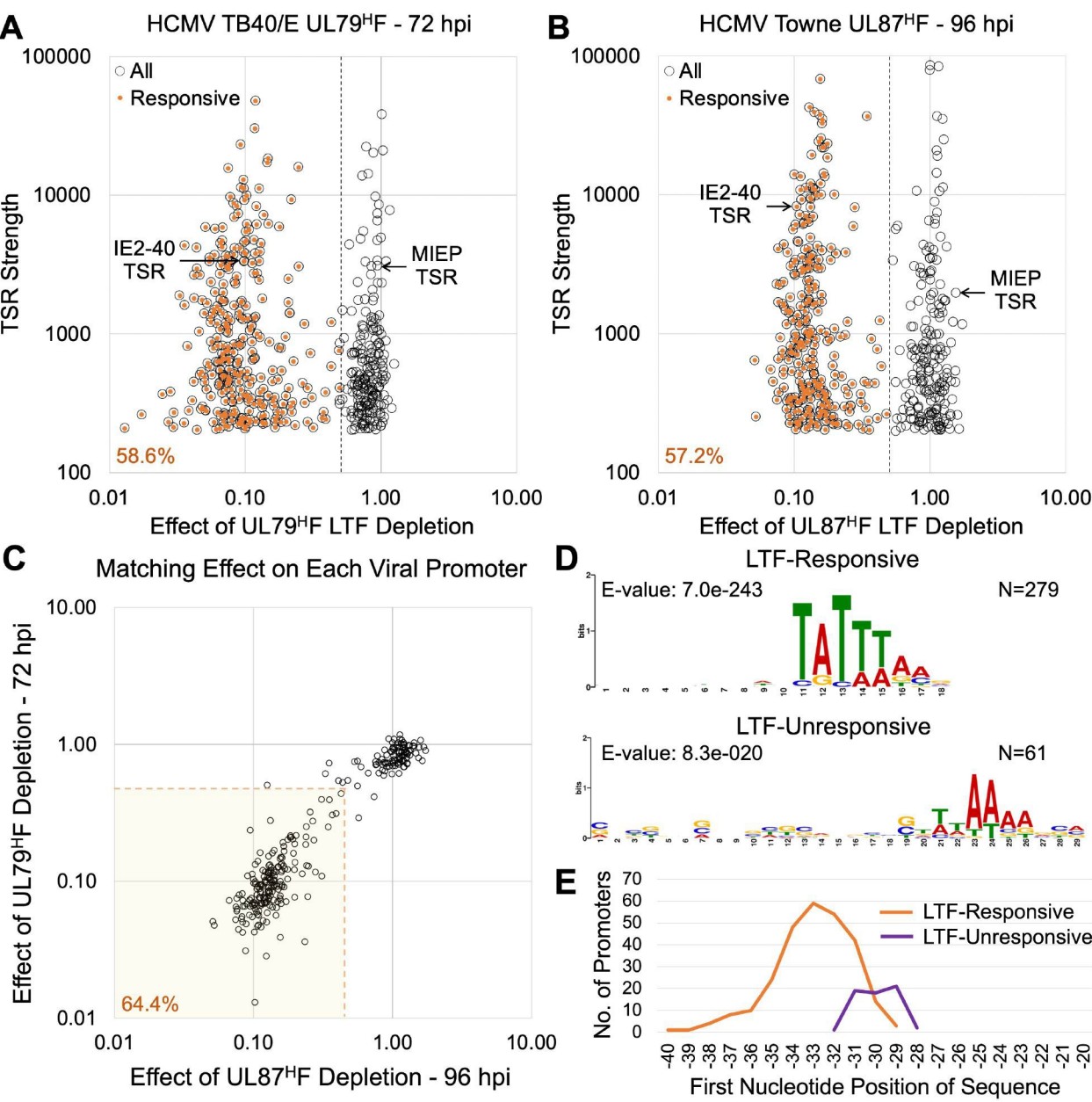

**Fig 3. Genome-wide characterization of LTF-responsive viral TSRs.** (**A and B**) HFF infected with HCMV TB40/E UL79HF (A) or Towne UL87HF (B) for 72 or 96 h, respectively, were treated with dTag2 or dTag1 for the last 6 h of infection. TSR strength is the number of Pol II nascent RNAs originating within the 20-bp transcription start region. Number of nascent RNA reads at each HCMV TSR were quantified by PRO-Seq-Flavo method. Viral TSRs of >200 reads in CTRL group (open circles) were plotted against the change in TSR strength for dTag vs CTRL treatment (dTag/CTRL). LTF-activated TSRs (orange dots) represent TSRs decreasing in strength by more than 50% because of dTag treatment (LTF-activated TSRs). Dashed lines denote 0.5 dTag/CTRL read breakpoints. (**C**) Comparison of TB40/E UL79HF LTF (72 hpi) vs Towne UL87HF LTF (96 hpi) depletion outcomes for 302 viral TSRs (>200 reads in CTRL) that are conserved between viruses. Shaded area contains the viral TSRs that are responsive to both UL79HF and UL87HF LTFs, defined as dTag/CTRL reads <0.5. Dashed lines denote 0.5 dTag/CTRL read breakpoints. (**D**) MEME motif discovery in the region -50 to -20 upstream of the MAXTSS of UL87HF LTF-responsive (dTag/CTRL reads <0.5) and -unresponsive viral TSRs. Logos represent MEME algorithm matches for 279 of 279 LTF-responsive TSRs and 61 of 178 LTF-unresponsive TSRs. (**E**) Positions of logo sequences relative to MAXTSS of UL87HF LTF-responsive (N = 268) and -unresponsive (N = 61) promoters, using panel B dataset.

T) prominence that is not absolute (**Fig 3D**). We verified that the first nucleotide in this logo is positioned -40 to -29 upstream of the MAXTSS for LTF-responsive TSRs, with greatest predilection for the -33 position (**Fig 3E**). This is consistent with previous findings from

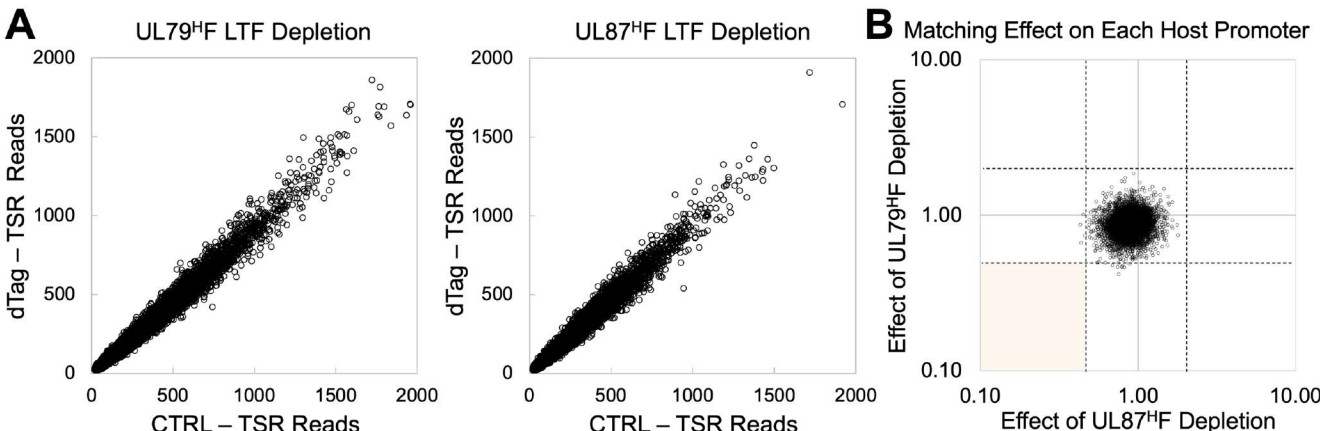

**Fig 4. Host promoter transcription is unaffected by viral LTF depletion.** (**A**) Effects of UL79HF or UL87HF LTF depletion (6-h dTag treatment) at 72 hpi on 10954 and 10497 host TSRs, respectively. Results derived from spike-in normalized PRO-Seq-Flavo datasets comparing dTag vs. CTRL reads at host TSRs having >20 CTRL reads. The same datasets were used in **Fig 3A and 3C** (UL79HF depletion at 72 hpi) and **S4A and S4B Fig** (UL87HF LTF depletion at 72 hpi). (**B**) Comparison of TB40/E UL79HF LTF vs Towne UL87HF LTF depletion results for 10,215 active host TSRs using PRO-Seq-Flavo datasets applied in **Fig 3A and 3C** (UL79HF depletion at 72 hpi)) and **S4A and S4B Fig** (UL87HF LTF depletion at 72 hpi). Each open circle depicts a host TSR >20 CTRL reads in both Towne UL87HF vs TB40/E UL79HF datasets. Dashed lines denote breakpoints at which dTag/CTRL reads are 0.5 or 2. Light gold area is where host TSRs affected by both LTFs with dTag/CTRL read ratio <0.5 would be located.

PRO-Cap-based analysis of TATT-containing viral promoters [29]. For LTF-unresponsive TSRs, the MEME algorithm yields an indistinct TA-rich logo sequence for 61 of the 178 LTF-unresponsive promoters (**Fig 3D**), with the first T positioned closer to the MAXTSS than is the TATT (**Fig 3E**). Many LTF-unresponsive promoter TSRs do not have a TATA or TATT. Several of these viral promoters have been previously shown to be activated by viral IE2 during late infection [26,29].

We next analyzed the same PRO-Seq-Flavo datasets to determine if LTFs affect host transcription at 72 hpi. Because the read depth at individual host genes is orders of magnitude less than that of virus genes, all host TSRs producing >20 reads in CTRL-treated infections were deemed to be evaluable host promoters that were active during the presence of LTFs. Analyses of 6-h dTag versus CTRL treatment effects on host transcription were applied to HCMV TB40/E UL79HF and Towne UL87HF infections that were analyzed for treatment effects on viral transcription, with the corresponding results shown in **Figs 3 and S4**. Only a few out of over 10,000 host promoter TSRs had TSR strength decrease by >2-fold because of UL79 or UL87 LTF depletion (**Fig 4A**). These host TSR outliers marginally exceeded the 2-fold cutoff and none of them were observed to be concordantly affected by both UL79 and UL87 LTFs (**Fig 4B**). Thus, viral LTFs drive transcription from promoters in the viral genome but not in the host genome.

## Promoter strength and the TATTW-containing octanucleotide sequence

The TATTW (W = A or T) is prominently featured in the 8-nucleotide MEME logo for all LTF-responsive viral TSRs (**Fig 3D**). The less enriched 3'-portion of this logo is more varied in nucleotide sequence. The amount of nascent RNA output at individual TSRs ranges over 2 orders of magnitude. To evaluate if TSR strength is linked to sequence variation in the TATTW-containing octanucleotide block, we first identified the UL87 LTF-responsive TSRs having a TATT positioned -40 to -28 from the MAXTSS (**Fig 5A**). 145 LTF-responsive TSRs were found to fit this criterion. We then ranked all LTF-responsive TSRs according to tertile of TSR strength and stratified TATT-containing promoters by this tertile rank. We found the

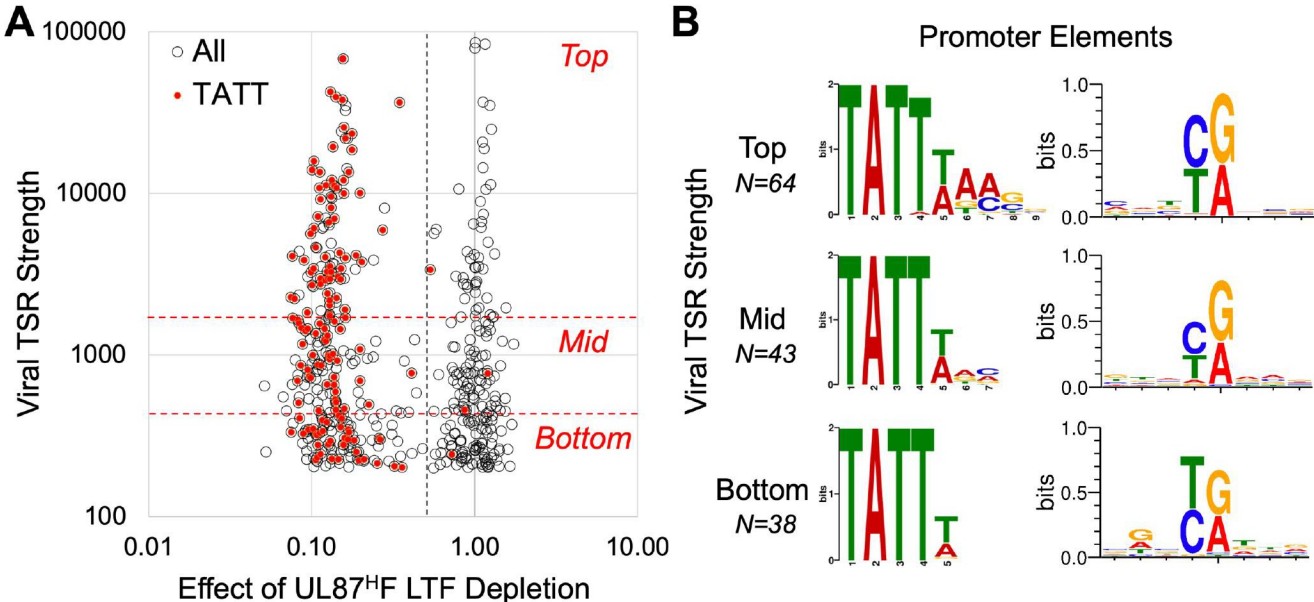

**Fig 5. TATTW-containing octanucleotide sequence in relation to strength of viral promoter transcription.** (**A**) Scatterplot of viral TSR strength (number of nascent RNA reads) versus effect of UL87<sup>H</sup>F LTF depletion on TSR strength (dTag vs. CTRL treatments) for all active viral TSRs (open circles). UL87 LTF-responsive TSRs divided into top, mid, and bottom tertiles based on individual-level viral TSR strength (red hatched lines). Viral TSRs having a TATT positioned -40 to -28 upstream of the MAXTSS are marked in red. Data derived from the experiment and method of measurement described in **Fig 3B**. (**B**) Analysis of core promoter elements by tertile rank in TSR strength. MEME algorithm was applied to positions -50 to -20 relative to MAXTSS for UL87 LTF-responsive TSRs having an upstream TATT. WebLogo was applied to positions -4 to +4 relative to MAXTSS.

TATT in the upstream sequence of 69%, 46%, and 41% of LTF-responsive TSRs ranking in top, mid, and bottom tertiles of TSR strength, respectively. The MEME algorithm was applied to determine if the logo sequence differed between TATT-containing promoters ranking in top and bottom tertiles of TSR strength (**Fig 5B**). This revealed that TATT-containing promoters in the top tertile are enriched in TATTWAMS (M = A or C and S = C or G), whereas promoters in the bottom tertile have the TATTW but lack enrichment of A at octanucleotide positions 6 and 7. The WebLogo tool was applied to demonstrate that the initiator sequence in these promoters does not appreciably differ in relation to tertile of TSR strength (**Fig 5B**).

The TATTWAA has been previously shown to confer LTF-dependent transcription in gammaherpesviruses [21,22]. We found that this sequence accounted for 23% of all HCMV UL87 LTF-responsive promoter TSRs having a TATT motif. The TATTTAA-containing promoters were disproportionately represented in the top tertile of TSR strength compared to TATTAAA promoters (Chi-square, p< 0.001). This was reflected in 12 of the 13 TATTTAA promoters ranking in the top tertile of TSR strength (median and mean TSR strength of 6,958 and 10,179 reads, respectively), whereas 11 of 21 TATTAAA promoters ranked in mid and bottom tertiles of TSR strength (median and mean TSR strength of 1,356 and 5,528 reads, respectively). Two promoters having the same 10-base TATTAAAGGT motif positioned -31 from the MAXTSS but differing 12-fold in TSR strength (2744 vs 224 reads) adds to the notion that other sequences or factors are contributing to TSR strength. In the case of these 2 promoters, neither are driven by viral IE2 in late infection [26]. By stratifying promoters by TSR strength and applying MEME differential enrichment mode, we were unable to discern a nucleotide motif extending downstream from the octanucleotide block that was linked to LTF responsiveness or TSR strength. Top performing TATTAAA promoters may have G and C enrichment downstream of the octanucleotide block compared to TATTAAA promoters

functioning in the bottom tertile. We have not discounted the possibility that nucleotide code in flanking sequence is a factor that modifies TSR strength.

Approximately 66% of LTF-responsive TATT-containing promoters ranking in the top tertile of TSR strength did not have the TATTWAA. This group of promoters still had the TATTW core. While a TATTTTA was in 3 top performing promoters (e.g., UL148 and US27 promoters), the TATTTT was significantly associated with lower TSR strength for the 16 TATTTT promoters ranking in mid and bottom tertiles (Chi-square, p< 0.001). Multiple top performing LTF-responsive promoters contained TATTTACS (e.g., UL47, UL52, UL72, UL74, UL89, and US17 promoters) or TATTAACG (e.g., UL73, UL99, UL119, and UL121 promoters) motifs. Other notable sequence variants include TATTTGCG (UL82 promoter), TATTAGAC (UL86 promoter), and TATTAAGG (UL94 promoter). The functional importance of the TATTW core is suggested by the observation that TSR strength decreases by almost 70-fold in association with a naturally occurring base-substitution in TATTTAA of the UL1 gene promoter in the Towne strain genome that converts this sequence to TATcTAA in the UL1 promoter in TB40/E strain genome. Taken together, the findings indicate that a limited degree of DNA sequence variation broadens the range of LTF targets and likely modulates individual-level effect on promoter transcription.

## LTFs control viral early-late and late promoters

Herpesvirus promoters are classified as true late promoters if they produce RNA only after onset of viral DNA replication. Viral early-late promoters produce RNA before onset of viral DNA replication and substantially more of the RNA after DNA replication. We applied two approaches and the PRO-Seq-Flavo method of measurement to determine the extent to which LTFs drive transcription from early-late versus late kinetic class promoters. We show in **Fig 6A** that UL79 LTF drives nascent RNA production from viral promoters for essential UL84 and UL85 genes. Both UL84 and UL85 promoters have a standard TATTTAA suitably positioned upstream of their MAXTSS. Time-series analysis of HCMV infection reveals activity of the UL84 promoter TSS before the onset of viral DNA replication at 12 hpi that gives rise to a 13-fold increase in UL84 TSR read density after onset of DNA replication at 48 hpi (**Fig 6B and 6C**). This temporal pattern of transcription fits with the early-late kinetics of UL84 RNA expression previously determined by northern blot [30]. In contrast, the UL85 promoter TSS is inactive before onset of viral DNA replication and becomes active once viral DNA replication has begun (**Fig 6B and 6C**). Applying phosphonoformic acid (PFA), a potent inhibitor of the viral DNA polymerase, throughout the 48-h infection lowered UL84 TSS use and almost completely prevented UL85 TSS use (**Fig 6B**). These two different outcomes are typical for early-late and late kinetic-class promoters, respectively, if PFA treatment is begun before onset of viral DNA replication. In contrast, applying the PFA treatment for 6 h in late infection had negligible effect on viral transcription (https://github.com/meierjl/Towne-UL87LTF), which is consistent with results of a prior study [26].

The metric of PFA sensitivity index (PSI) [26] was applied to determine the size and composition of the LTF-responsive early-late viral promoter population. PSI is computed as the ratio of promoter's TSR reads remaining after 72-h PFA treatment divided by TSR reads without PFA. The PSI values for all active viral TSRs at 72 hpi that rank in top and mid tertiles of TSR strength were plotted against the TSR strength change caused by UL79 LTF depletion (**Fig 6D**). Viral TSRs scoring PSI ≤45 typically have a TSS that is active before onset of viral DNA replication, as evident from 5'-end read alignments of spike-in normalized PRO-Seq-Flavo results for the HCMV infection time course. Nearly one-third of the viral TSRs at 72 hpi have PSI ≤45 and the UL79 LTF drives 47% of these TSRs. The UL79 LTF drives 98% of viral

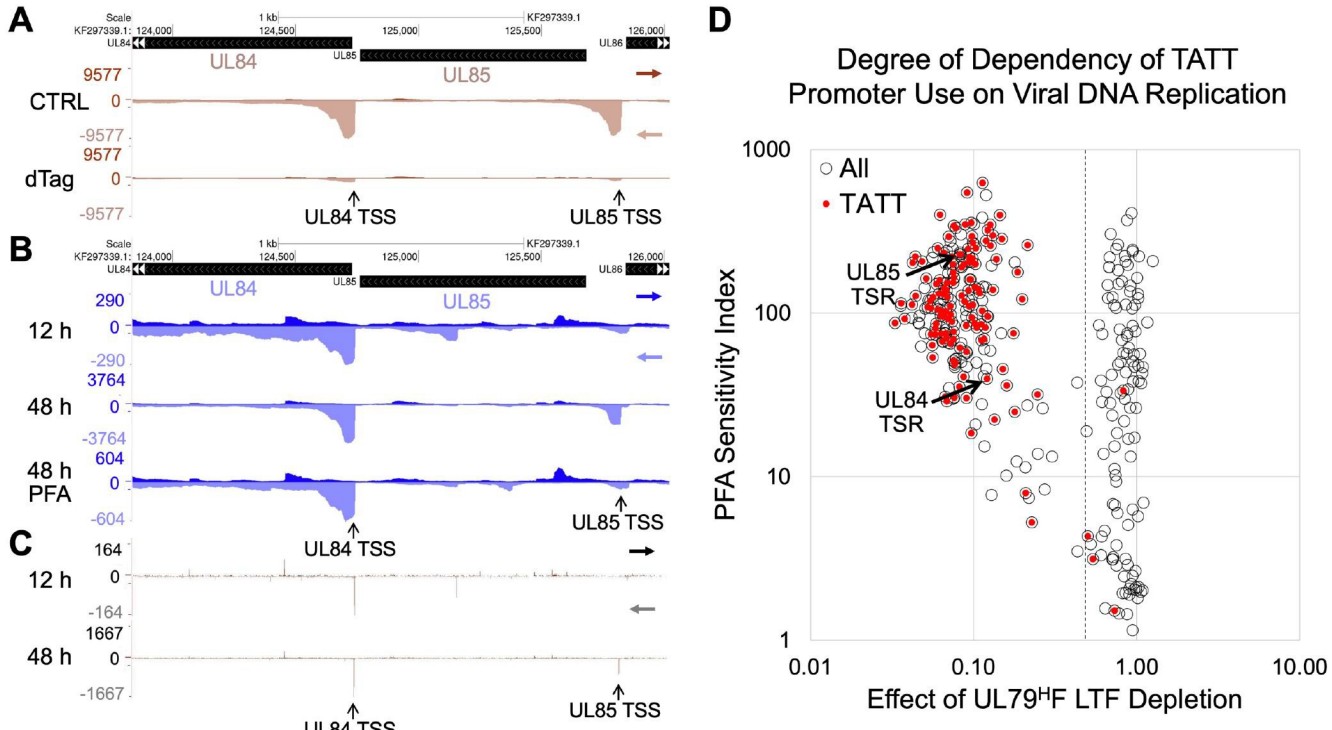

**Fig 6. The UL79 LTF initiates transcription at TATT-containing viral early-late and late promoters.** (A-C) Genome browser views of PRO-Seq-Flavo results for TATTTAA-containing UL84 and UL85 promoters. Spike-in normalized reads from Pol II nascent transcripts are aligned to the HCMV TB40/E KF297339 genome. (**A**) HCMV TB40/E UL79$^H$F infection for 72 h treated with 6-h dTag vs. CTRL. (**B** and **C**) HCMV TB40/E WT infections for 12 and 48 h (in absence or presence of PFA). Panel C is a map of 5'-ends of the same reads aligned in panel B for 12-h and 48-h infections without PFA. Vertical arrows point to positions of UL84 and UL85 MAXTSS. (**D**) Scatterplot of PFA sensitivity index (PSI) vs. effect of UL79$^H$F LTF depletion on TSR strength (dTag vs. CTRL treatments) for 320 viral TSRs ≥435 nascent RNA reads (open circles). Viral TSRs having a TATT positioned -40 to -28 upstream of the MAXTSS are marked in red. PSI = ratio of TSR reads for PFA treatment/CTRL treatment of 72-h HCMV TB40/E WT infection. Effect of UL79$^H$F LTF depletion (dTag/CTRL reads) is described in Fig 3B. Dashed line denotes 0.5 dTag/CTRL read breakpoint. Arrows point to UL84 and UL85 TSRs, with PSIs 40 and 220, respectively.

TSRs with a TATT positioned -40 to -28 bases from the MAXTSS. The PSI values for these promoters range widely from 5 to 626, with a median of 119. The early-late UL84 and late UL85 promoters score PSIs 40 and 220, respectively. The UL87 LTF-driven TSRs having a suitably positioned TATT also have PSIs ranging widely from 3 to 587, with a median PSI 105 (**S5A Fig**). Of the UL79 LTF-driven promoters having a TATT, 15% have TSRs with PSI ≤45. Only 4 TATT-containing promoters are unresponsive to UL79 LTF depletion, and 2 of these promoters were previously shown to be driven by viral IE2 in late infection [26]. We conclude that viral LTFs drive transcription from a sizable portion of the viral early-late promoter population and a subset of these promoters have the canonical TATT sequence.

## LTFs also act through noncanonical sequences

We were struck by the unanticipated finding that a substantial proportion of LTF-responsive viral TSRs did not have a TATT. Analysis of sequences suitably positioned upstream of the viral MAXTSS of TATT-less promoters uncovered four different patterns of noncanonical sequences. We grouped these sequences into categories of TATAT, TGTT, YRYT (Y = C or T; and R = A or G), and TATAA. The proportional distribution of UL79 LTF-driven promoters with noncanonical and canonical sequences varies in relation to tertile of TSR strength (**Fig 7A**). The proportion of promoters with a TGTT, YRYT, or TATAA motif is substantially

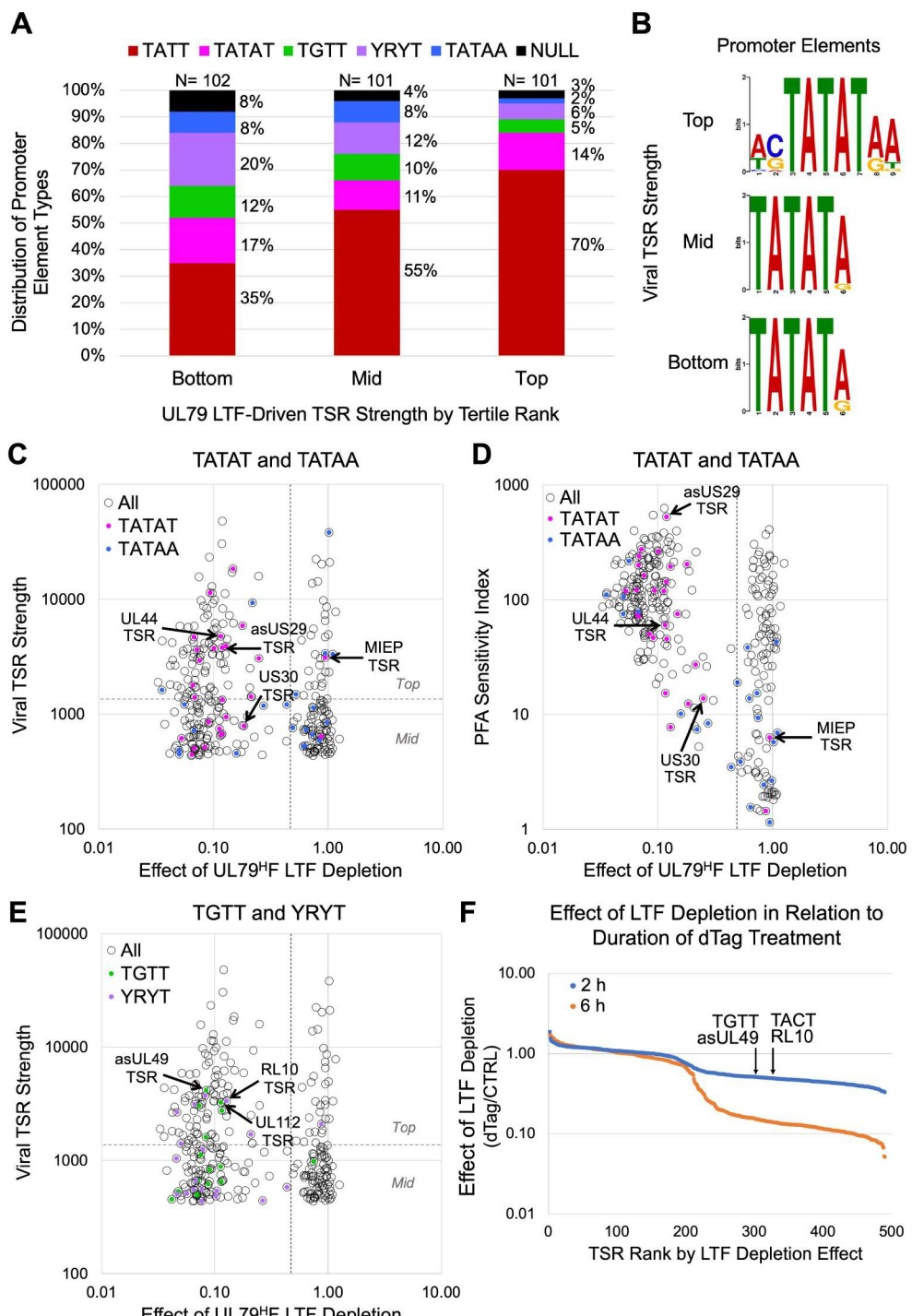

**Fig 7. Noncanonical sequences in LTF-responsive promoters.** (**A**) Proportional distribution of different motif types in LTF-responsive promoters by tertile rank in TSR strength. Based on effects of UL79HF LTF depletion (6-h dTag treatment) on viral TSR strength at 72 hpi. Null, unclassified sequence. (**B**) MEME logo determination for UL87 LTF-responsive TSRs having a TATAT positioned -50 to -20 relative to MAXTSS by tertile rank in TSR strength. (**C-E**) Effect of UL79HF LTF depletion on viral TSR strength at 72 hpi versus baseline TSR strength (C and E) or PSI (D) for viral TSRs ≥435 nascent RNA reads (open circles). Viral TSRs with TATT (red), TATAT (pink), TATAA (blue), TGTT (green), or YRYT (violet) sequences positioned -40 to -28 relative to MAXTSS are marked with dots colored accordingly. Horizontal dashed line divides top and mid TSR strength tertiles. (**F**) Effect of dTag treatment for 2 versus 6 h on viral TSR strength (dTag/CTRL) at 96 hpi for HCMV UL87HF. Viral TSR strength was determined by spike-in

normalized PRO-Seq-Flavo method. Arrows point to viral asUL49 and RL10 TSRs having upstream TGTT and TACT sequences, respectively.

larger in the bottom than in the top tertile and the proportion of promoters with TATT is substantially larger in the top than in the bottom tertile (Chi-square, p< 0.001). In contrast, the proportion of promoters with a TATAT differs little across tertiles. The TATAA was the least likely of the 4 noncanonical motif types to be represented in LTF-responsive promoters and mostly relegated to promoters in mid and bottom tertiles. Notably, the proportion of LTF-responsive promoters with an unclassified sequence decreased as tertile rank in TSR strength increased (8%, 5%, and 3% in bottom, mid, and top tertiles, respectively). This observation suggests that LTF-mediated effects are less sequence-specific or indirect for viral TSRs producing a low amount of nascent RNA reads.

The TATAT is present in nearly half of all noncanonical promoters ranking in the top tertile of TSR strength. MEME analysis revealed that TATAT flanking sequence differed by tertile rank (**Fig 7B**). It also picked up on the TATATA palindrome that had been identified by manual inspection of sequences upstream of all TSRs producing >200 unique nascent RNA reads. We had identified 17 TATATA palindromes in the HCMV genome that were linked to LTF-driven transcription. Six of these palindromes were linked to bidirectional transcription from back-to-back promoters, in which the palindrome was positioned at -35 to -31 from each MAXTSS in the promoter pair. In all these cases, TSR strength differed greatly between the promoters in the pair. The TATATAA sequence pattern was favored in stronger promoters. This is reflected in the robustness of the LTF-responsive UL44 late promoter TSR, which was previously known to have a TATATAA motif positioned upstream [31] (**Figs 7C and** S5B). In contrast, the divergent asUL45 promoter TSR is in the bottom tertile of TSR strength and linked to a TATATGA. A TATATA was also found to be positioned between divergent asUL29 and US30 promoters having TSRs ranking in top and mid tertiles of TSR strength, respectively (**Fig 7C**). The TATATAA motif is linked to the stronger asUS29 TSR, whereas a TATATAC is linked to the weaker US30 TSR. The asUS29 and US30 promoters were distinguished as late and early-late promoters, respectively, based on PSI scores (PSI 526 and 14, respectively) (**Figs 7D and** 5C) and time series analysis of transcription before and after viral DNA replication. Surprisingly, the MIEP TATA box has the TATATAA motif, yet the depletion of either UL79 or UL87 LTF had not changed MIEP TSR strength (**Figs 7C and** S5B). Other factors may be involved in preventing LFTs from acting on this TATATAA.

One in 10 LTF-responsive promoters ranking in the top tertile of TSR strength had an upstream TGTT or YRYT sequence (**Fig 7A**) and most of these TSRs had PSI values in the range of true late promoters (**S6A Fig**). The CTGTTTAM (M = A or C) motif was mostly represented in top performing TGTT-promoters, when compared to mid and bottom tertiles (Fisher's exact test, p = 0.035). Examples include the CTGTTTAA positioned -30 and -32 from the MAXTSS of asUL49 and UL112 gene promoters, respectively (**Fig 7E**). The TACTACA positioned -31 from the MAXTSS for the robust TSR of the RL10 gene promoter is an example of a YRYT motif that is linked to LTF-driven transcription (**Fig 7E**). Results of analyses of noncanonical sequences upstream of UL87 LTF-responsive TSRs corroborate the findings of the UL79 LTF studies (**S6B Fig**). An indirect effect is unlikely to explain these results because shortening the timeframe of LTF depletion to 2 h lowered transcription from LTF-responsive noncanonical and canonical promoters to a comparable degree, although 6 h of LTF depletion was more effective than 2 h of LTF depletion in lowering transcription (**Fig 7F**). Also, blocking viral DNA synthesis for 6 h in late infection (with or without concomitant 2-h LTF depletion) had no effect on viral transcription (https://github.com/meierjl/Towne-UL87LTF), supporting

the assertion that changes in transcription caused by short-term LTF depletion are not due to indirect effects on viral DNA replication. The totality of findings indicates that LTFs engage a limited range of different sequence motif blocks with sequence patterns that correlate with promoter strength.

## LTFs directly target noncanonical and canonical sequences

We applied ChIP-Seq to determine if LTFs occupy noncanonical sequences in LTF-driven viral promoters in late infection, as well as occupy canonical sequences like those previously described for canonical KSHV TATTW promoters [23]. Formaldehyde crosslinked chromatin isolated from HFF infected with HCMVs expressing HA-tagged UL79 and UL87 LTFs was sheared by sonication, immunoprecipitated with anti-HA or anti-Pol II antibody, and the resultant DNA fragments <400-bp in length subjected to paired-end sequencing. Genome browser views of de-duplicated sequencing reads aligned to the respective genomes of two different HCMV strains reveal a similar distribution of UL79 and UL87 LTF ChIP-Seq peaks (**Fig 8A**). Inspection of the LTF-unresponsive UL83 promoter revealed that it differs from neighboring LTF-responsive UL80.5 and UL82 promoters in lacking UL79 and UL87 LTF ChIP-Seq peaks despite having a prominent paused Pol II ChIP-Seq peak (**Fig 8B**). This finding is consonant with prior evidence that viral IE2 and not LTFs drive UL83 promoter transcription [26] and speaks to the accuracy of the LTF ChIP-Seq results. We next focused attention on LTF-responsive noncanonical promoters. To help identify LTF binding sites underneath broad ChIP-Seq peaks, the reads for DNA fragments were sorted by fragment length and resulted in a grayscale heatmap of fragment count by fragment length. The heatmap produces a dark triangular silhouette with its vertex corresponding to the area of LTF occupancy. As shown in **Fig 8C**, the UL79 and UL87 LTF ChIP-Seq footprint patterns are virtually identical in the viral genome region carrying UL48A and asUL49 promoters. LTF occupancy is substantial at the noncanonical asUL49 promoter containing the CTGTTTA, as well as the canonical UL48A promoter containing the TATTATC variant. LTFs also occupy the 60-bp region between back-to-back LTF-responsive asUS29 and US30 promoters (**Fig 8D**). An exploded view of this region reveals UL87 and UL79 LTF ChIP-Seq read peaks with vertices positioned slightly left of center between the TSRs for these promoters. This slight leftward shift may reflect TATATA palindrome positioning that is slightly closer to the asUS29 MAXTSS than it is to the US30 MAXTSS but might also reflect the >10-fold difference in asUS29 TSR strength over US30 TSR strength. Nevertheless, both LTFs bind to the small DNA region shared by asUS29a and US30 promoters that contains the TATATA palindrome and has no TATT sequence.

We evaluated whether the amount of UL87 LTF ChIP-Seq signal corresponds to the amount of viral TSR strength at 72 hpi. To increase the signal-to-background noise ratio for measuring UL87 LTF ChIP-Seq peak size, the midpoint (center) of each UL87 LTF ChIP-Seq fragment of 100–200 bp length, termed Frag Center, was mapped to the HCMV Towne genome. An example of aligned Frag Centers to the viral genome region containing the MIEP and downstream promoters is shown in comparison to a standard alignment of all UL87 LTF ChIP-Seq fragments, termed Total Frags, which range from 18 to 400 bp in length (**Fig 9A**). In this region of the viral genome, the levels of occupancy of the UL87 LTF-responsive promoters by the viral UL87 LTF appear to be comparable to levels of the TSR strength for these promoters, as measured by the PRO-Seq-Flavo method. UL87 LTF ChIP-Seq peaks are absent in the LTF-unresponsive MIEP and enhancer promoter. To gauge the size of a UL87 LTF ChIP-Seq peak in an LTF-responsive promoter, we counted Frag Centers located within a 200 bp window centered on base position -25 relative to the MAXTSS of the promoter's TSR. By evaluation of all possible UL87 LTF ChIP-Seq peaks distributed across the viral genome, it was

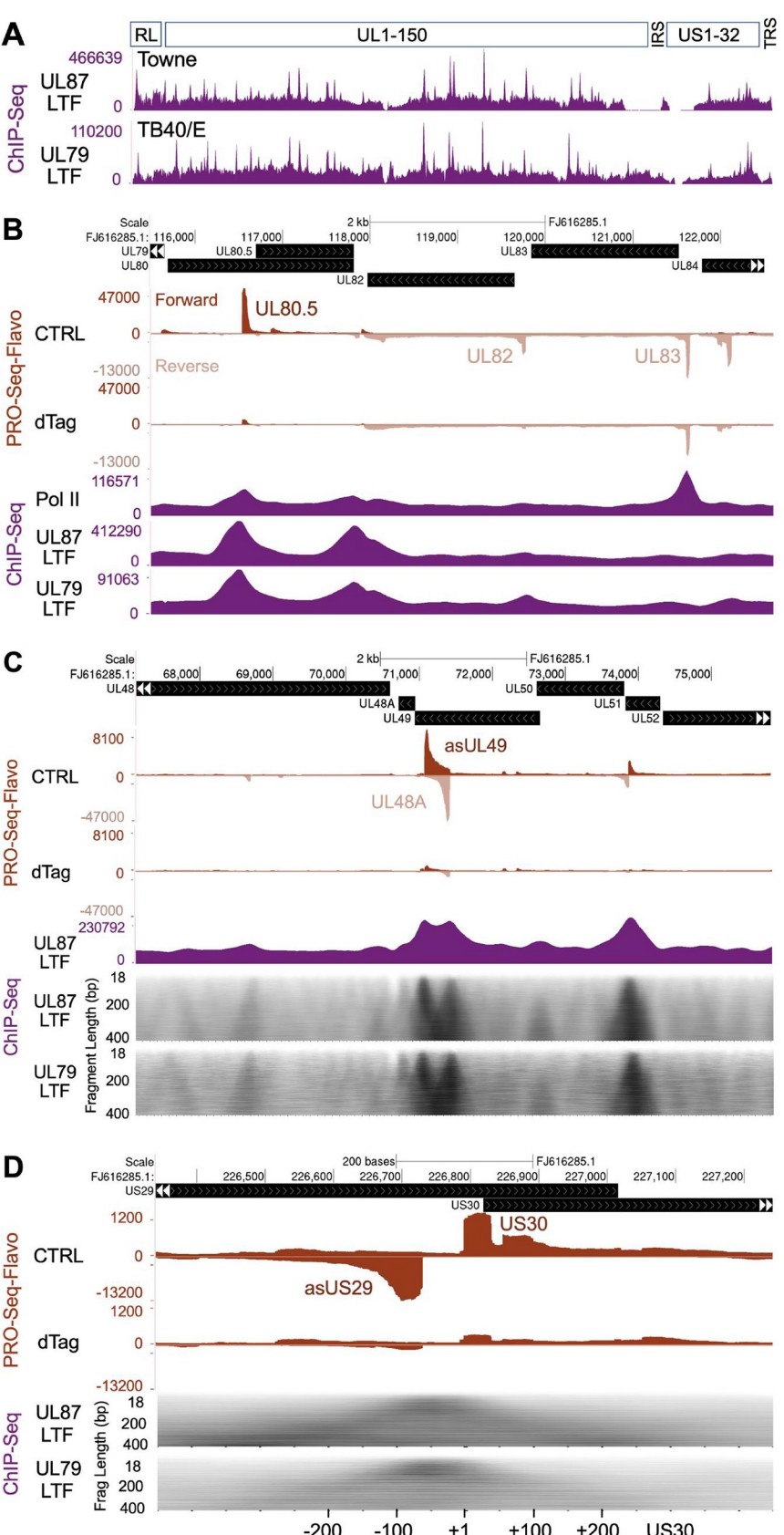

**Fig 8. Viral genome sites of LTF occupancy.** (**A**) HCMV genome-wide view of ChIP-Seq results for tagged UL87 and UL79 LTFs at 72 hpi (MOI of 3) expressed by Towne and TB40/E strains, respectively. UL87 and UL79 LTF ChIP-Seq reads were aligned to Towne (FJ616285.1) and TB40/E (KF297339.1) genomes, respectively. Uniquely mapped sequence reads after removal of PCR duplicates constituted DNA fragments 18–400 bp in length. The pileups of these fragments are displayed in spatial relationship to schematically depicted HCMV genome regions—repeat long, RL; unique long, UL; internal repeat short, IRS; unique short, US; and terminal repeat short, TRS. (**B**) Pol II and LTF ChIP-Seq results compared to spike-in normalized PRO-Seq-Flavo results for 6-h dTag vs. CTRL treatments of HCMV Towne UL87$^{HF}$-infected HFF, which is described in Fig 2. UCSC Genome browser views of aligned PRO-Seq-Flavo reads were scaled to allow comparison of dTag vs. CTRL results. Locations of the LTF-unresponsive UL83 promoter TSR and LTF-responsive UL80.5 and UL82 promoter TSRs are denoted. Pol II and UL87 LTF ChIP-Seq were performed in parallel, and the sequence reads were aligned to Towne genome; UL79 LTF ChIP-Seq reads were aligned to TB40/E genome. (**C**) Browser views of LTF-responsive asUL49 CTGTTTAA and UL48A TATTATC promoters comparing LTF ChIP-Seq and PRO-Seq-Flavo results, using approaches described in panel B. Gray scale heatmaps of UL79 and UL87 LTF ChIP-Seq fragment quantity vs. length (18–400 bp) better resolve positions of UL79 and UL87 LTF ChIP-Seq peak summits mapped to TB40/E and Towne genomes, respectively. (**D**) Exploded view of asUS29 and US30 promoters showing LTF occupancy in relation to transcription, using the same datasets and approaches applied in panels B and C. The TATATA is positioned -32 and -34 relative to asUS29 MAXTSS and US30 MAXTSS, respectively. The browser view is scaled to allow viewing of both the small US30 TSR and the large asUS29 TSR.

determined that Frag Center signal counts >15000 provided evaluable UL87 ChIP-Seq peak signals, whereas the peak signals below this cutoff were controverted by the background signals. We excluded from analysis the ChIP-Seq peaks over palindromes that control two promoters and the overlapping peaks belonging to more than one promoter. This provided 73 evaluable UL87 LTF ChIP-Seq peaks for which signal counts were compared with the TSR strength of these promoters. The level of TSR strength was determined from the PRO-Seq-Flavo dataset labeled CTRL in **Fig 9A**. The plot (**Fig 9B**) of the comparison of amount of UL87 LTF ChIP-Seq peak signals vs. TSR strength reveals a degree of correlation that implies the UL87 LTF occupancy level is predictive of the strength of LTF-driven promoter transcription. Of these 73 evaluable LTF UL87 ChIP-Seq peaks, ~89% of them are in top performing viral promoters for TSR strength, whereas the remaining 11% of the ChIP-Seq peaks are in high mid-level performing promoters. TATT-less promoters account for 29% of the evaluable UL87 LTF ChIP-Seq peaks in LTF-responsive promoters. There are several possible reasons why the correlation between levels of ChIP-Seq signal and TSR strength is not strictly one-to-one, including the variable of deoxynucleotide composition itself as a determinant of protein-DNA crosslinking efficiency by formaldehyde [32,33].

## Discussion

Here we describe a previously unknown constellation of findings that unveil a new view of herpesvirus LTF function that expands the range of viral promoter targets and likely modulates individual promoter output. The discovery owes to the integration of recent technological advances that enable the identification and characterization of LTF-responsive and -unresponsive promoter populations, quantification of nascent transcripts at individual promoters, precision mapping of when and where Pol II initiates transcription, and genome-wide identification of sites of LTF binding. The combined study results provide compelling evidence in late HCMV infection that UL79 and UL87 LTFs bring about transcription from the same set of viral promoters and occupy the same DNA sequence types in these promoters. The idea of UL79 and UL87 LTFs functioning together to initiate transcription aligns with prior reports indicating that UL79 and UL87 LTFs co-immunoprecipitate from lysates of HCMV-infected fibroblasts [6], as well as results of other protein-protein interaction studies for comparable LTF homologs of murine CMV [8]. To examine LTF-mediated effects, we deplete UL79 and UL87 LTFs over a short timeframe to minimize risk of indirect effects changing transcription. This might explain why we do not find evidence of the UL79 LTF functioning as a

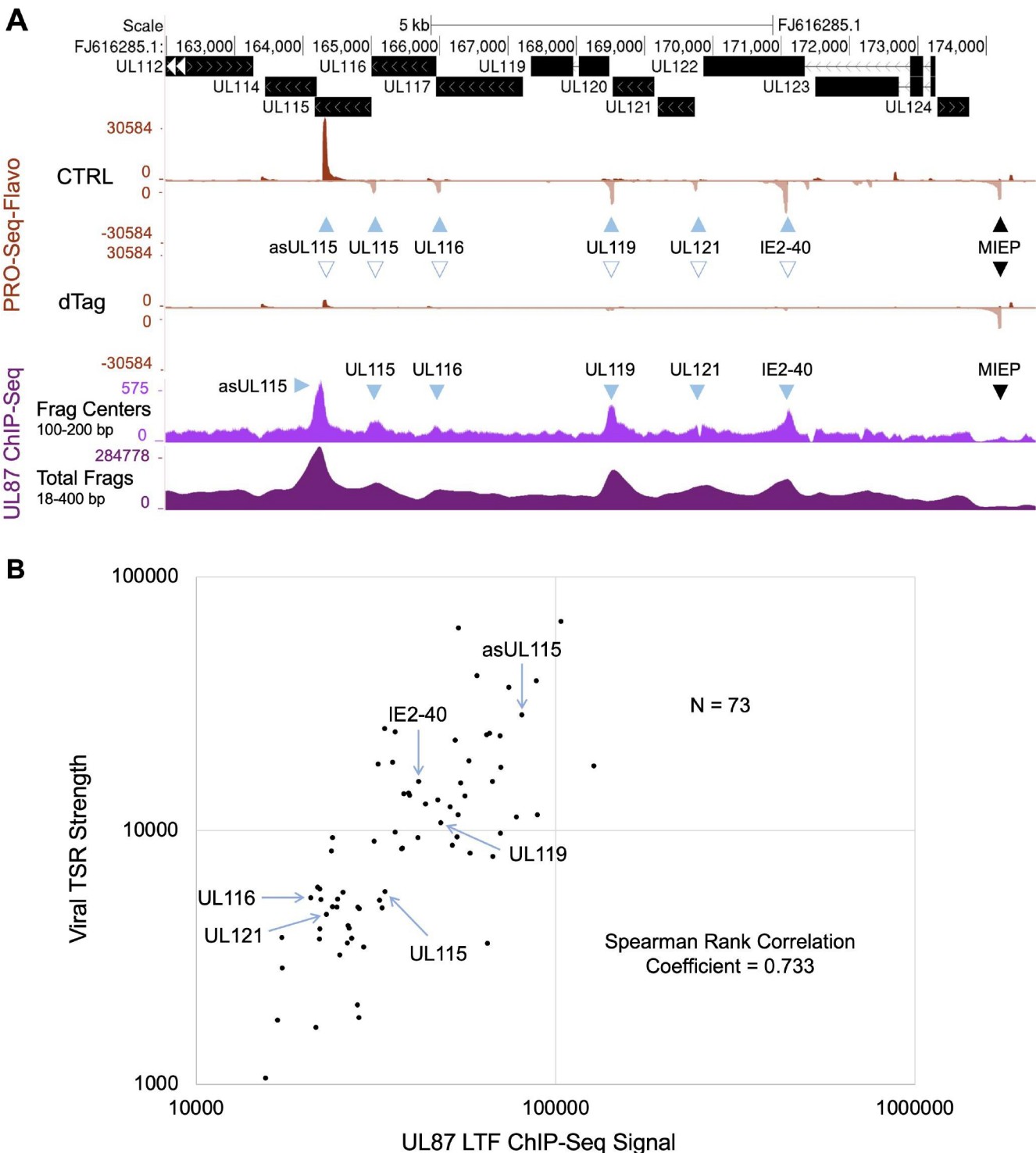

**Fig 9. Relationship of promoter TSR strength and occupancy by UL87 LTF at 72 hpi.** (**A**) UCSC Genome browser view of the distribution of the midpoints (centers) of UL87 LTF ChIP-Seq fragments of 100–200 bp in length (Frag Centers) mapped to the HCMV Towne MIEP and downstream promoters. This is compared to the distribution map of all UL87 LTF ChIP-Seq fragments (Total Frags), which range in length from 18 to 400 bp. PRO-Seq-Flavo analyses of vehicle CTRL vs. dTag treatment for the final 6 h of the infection reveal locations and sizes of UL87 LTF-responsive and -unresponsive promoter TSRs. Blue arrows point to LTF-responsive viral promoter TSRs with corresponding UL87 LTF ChIP-Seq peaks that have >15000 ChIP-Seq Frag Centers (in fragments of 100–200 bp length) located within a 200 bp window centered on base position -25 relative to the MAXTSS for the TSR. Black arrow denotes location of the LTF-unresponsive MIE promoter TSR. (**B**) Scatterplot shows the relationship of UL87 LTF ChIP-Seq signal and TSR strength. The Frag Center count within the 200 bp window described above represents the number of signals in the UL87 LTF ChIP-Seq peak. TSR strength is the number

of Pol II nascent RNAs originating within the 20-bp transcription start region. Blue arrows mark the LTF-responsive viral TSRs having UL87 LTF ChIP-Seq peaks of sufficient size that are pointed out in panel A. Seventy-three pairs of LTF-responsive viral TSRs and UL87 LTF ChIP-Seq peaks were deemed evaluable genome-wide, using inclusion and exclusion criteria described above and further detailed in the Results and Materials and Methods sections. Spearman rank correlation coefficient = 0.733.

transcription elongation factor, based on PRO-Seq analysis of infections carried out in the absence of Flavo. In a prior study, UL79 LTF depletion throughout infection decreased nuclear run-on RNA products across multiple viral genes without decreasing Pol II occupancy at the promoters of these genes, which included the IE1 gene and its MIEP [6]. Unlike the RT-PCR that was used to quantify the nuclear run-on RNA products, PRO-Seq obviates confounding by LTF-dependent antisense transcripts, such as those spanning the IE1 gene, or sense transcripts originating from unrecognized promoters downstream of the MIEP (**S3 Fig**). With regard to host transcription, we find that the 6-h depletion of UL79 and UL87 LTFs does not appreciably affect transcription from the 10,215 active cellular promoters. This observation supports findings of a prior meta-analysis of PRO-Cap data indicating that a TATT sequence suitably positioned in active cellular promoters does not increase transcription during late HCMV infection [29]. We have speculated that viral LTFs are unable to access the DNA in host chromatin [29]. Quantification and mapping of nascent transcripts coupled to engaged Pol II more accurately reflects how transcription changes in relation to stage of infection or LTF loss than does measurements of mature RNAs that are subject to posttranscriptional regulation. We took this tactic to draw clear distinctions between LTF-responsive and -unresponsive promoter populations and gauge individual level promoter transcription. This was key to recognizing the heterogeneity in the LTF-responsive promoter population and the bias in specific motif types or sequence variants for strong vs. weak promoters. Concordance of findings across different times in late infection (72 and 96 hpi) and different HCMV strains (Towne and TB40/E) speaks to reproducibility of the results and strengthens our conclusions. We further show that LTFs act directly on this diverse promoter population based on evidence of LTF binding to these promoters in infected cells and the immediate response of these different promoters to rapid LTF loss. Short-term LTF loss is not indirectly acting through an effect on viral DNA replication because parallel control studies demonstrate that potently inhibiting viral DNA synthesis for 6 h in late infection does not affect viral transcription.

While it is well known that herpesvirus late promoters are regulated by core promoter elements, the mechanisms regulating the strength of individual late promoters have been elusive. Nandakumar and Glaunsinger recently reported that for KSHV a 5-bp motif positioned immediately downstream of the TATTWAA contributes to LTF binding and promoter strength [22]. Our study methodology provides a snapshot in time of the quantity of Pol II nascent RNA reads at a promoter's MAXTSS (or TSR strength) that represents the frequency with which Pol II is engaged at the promoter among a population of genomes carrying this promoter. Analysis of PRO-Seq-Flavo findings for the infection condition shown in **Fig 2A** indicates that ~1 million HCMV genomes have Pol II engaged at the RNA4.9 promoter at 96 hpi. If we venture the assumption that 1 million viral genomes are amenable to Pol II transcription at other viral promoters, it will take 0.1–3.0% of such viral genomes at that point in time for Pol II engagement to yield TSR strength values of the top performing LTF-responsive promoters described in this report. An LTF-responsive promoter falling in the bottom tertile of TSR strength would be active on <0.03% of such viral genomes. We surmise that the LTF target sequence code determines probability of promoter activity based on several lines of circumstantial evidence. First, the canonical TATT motif is heavily biased toward the strong promoter group, whereas noncanonical motifs are heavily biased toward the weak promoter

group. Second, the TATTWAMS sequence pattern is distinctively featured in the strong canonical promoter group and not in the weak canonical promoter group. Third, TATAT promoters are more apt to be stronger if the sequence pattern is TATATAA, and TGTT promoters are stronger if the sequence pattern is CTGTTTAM. Fourth, individual promoter-level LTF-driven transcription correlates with the intensity of LTF ChIP-Seq peak signal at that promoter. Fifth, a naturally occurring base-substitution that changes TATTTAA to TATcTAA in the UL1 gene promoter is associated with ~70-fold decrease in transcription from this promoter during late infection. We looked but did not find evidence of initiator sequence differing between LTF-responsive promoters in top vs. bottom tertiles of transcription level. Nor did we find distance between TATT and MAXTSS to be associated with level of promoter output.

The TATTWAA is known to confer LTF-dependent transcription in gammaherpesviruses [21,22]. We find only 24% of LTF-responsive HCMV promoters ranking in the top tertile have the TATTWAA. The TATTTAA is almost exclusively featured in top performing LTF-responsive promoters, but only accounts for 13% of these promoters. Of the 21 LTF-driven promoters with a TATTAAA, only half of them rank in the top tertile of promoter strength and this top half appears to have G or C enrichment in downstream flanking sequence. This observation indicates that sequence code in the octanucleotide block is not the sole determinant of promoter strength. Through time series analyses and PSI determinations, we demonstrate the ability of HCMV LTFs to drive transcription from TATTWAA-containing early-late and late kinetic-class promoters, a finding that aligns with recent findings described for gammaherpesviruses [21,22]. TATTW variants that are not TATTWAA account for 24% of top performing promoters and generally but not always conform to the TATTWRMS. Deviation from the TATTWRMS is a common characteristic of weaker promoters that have the TATTW.

The discovery that many LTF-driven viral promoters lack the TATT was unexpected. Approximately 25%, 33%, and 49% of top, mid, and bottom performing promoters, respectively, that fit this category have a TATAT, TGTT, or YRYT. The TATAT is present in nearly half of the top performing noncanonical promoters. The TATAT had been previously implicated in LTF activation of gammaherpesvirus late promoters [22] and was shown to mediate LTF-driven gene expression in transient reporter assay [34,35]. Here, we uncover this motif's unique property as a palindromic variant shared by back-to-back promoters that is occupied by LTFs. Multiple promoter pairs of this kind are dispersed across the viral genome and ~60 bases separate the two MAXTSSs for each promoter pair. This type of divergent transcription differs from that observed during transcription of mammalian genes, which is coupled to upstream transcription in the opposite direction. Adding CTGTTTAA and TACTACA to the repertoire of LTF target sequences further expands the program of LTF regulated transcription. While the majority of LTF-dependent noncanonical promoters are true late promoters, early-late promoters are also represented. LTFs target at least 15% of the same viral promoters that are active in early infection. Many other viral genes have separate early and late promoters that are closely spaced, with the early promoter still active after onset of viral DNA replication but the late promoter only active after onset of viral DNA replication and dependent on LTFs. Thus, diversity in LTF target sequences shapes the LTF-transformative program that drives the viral early-to-late transcription switch.

## Materials and methods

### Cells and viruses

De-identified discarded human foreskins were used to isolate HFF, in accordance with IRB approval. HFF were propagated in Minimum Essential Medium (Gibco, 11095080)

supplemented with heat-inactivated 5% fetal bovine serum (Gibco, 26140079) and 1000 U/mL penicillin-streptomycin (Gibco, 15140122). HFF were studied at passage number ≤6. Construction and propagation of HCMV Towne UL87[H] and Towne UL87[H]F were previously reported by M. Li, et al [26]. TB40/E BAC4 was used to construct TB40/E UL79[H] and TB40/E UL79[H]F. Oligonucleotides and double-stranded DNA gene blocks that were used to construct TB40/E UL79[H] and TB40/E UL79[H]F viruses are listed in **S1 Table**. To construct HCMV TB40/E UL79[H], the bacterial galK gene was placed at the amino-terminus of UL79 ORF. The galK gene was then replaced by DNA encoding the HA epitope (YPYDVPDYA-L-protein) to fuse HA in-frame to the N-terminus of the UL79 ORF. HCMV TB40/E UL79[H]F was constructed by placing the bacterial galK gene at the carboxy-terminus of UL79[H]. The galK gene was replaced by a gBlock (Integrated DNA Technologies) containing FKBP12M. The FKBP12M was fused in-frame with the C-terminus of the UL79[H]. The region of the viral genome that was changed was analyzed by Sanger sequencing and the entire viral genome was subjected to restriction fragment length polymorphism analysis. HCMV BAC DNA was nucleofected into HFF using the Amaxa Neonatal Human Dermal Fibroblasts kit (Lonza, VPD-001) and the Amaxa Nucleofector II at program setting U23. These virally infected HFF were mixed with ARPE-19 cells, which is a human retinal pigmented epithelial cell line. Infected ARPE-19 were aliquoted, stored, and used to inoculate HFF to make a larger viral stock for experiments. The number of HCMV TB40/E replication cycles in HFF was minimized to preserve the phenotype-genotype of TB40/E recombinant viruses. Viral inoculums used in experiments were prepared from supernatant of infected HFF. The supernatant was passed through a 0.45 μm filter and virus in the filtered supernatant was pelleted through a 20% sorbitol cushion. Viral stocks used for comparing different viruses were tittered in parallel. To determine the viral titer of the inoculum (infectious units per cell), the inoculum was serially diluted and applied to a 24-well plate of HFF and examined by immunofluorescence assay of HCMV IE1/IE2 protein-positive cells at 24 h pi, using murine monoclonal antibody MAB810 (EMD Millipore, 1:1000 dilution) followed by a secondary goat anti mouse IgG (H+L) antibody conjugated to Alexa Fluor 555 (Thermo Fisher Scientific, A-21422). DAPI counterstain of nuclei enabled determination of percent of cells infected.

## PRO-Seq and ChIP-Seq experiments

T-150 cm$^2$ flasks of confluent HFF inoculated with HCMV at MOI of 3 were used for PRO-Seq or ChIP-Seq experiments. All infected HFF were maintained in the same growth medium detailed above. Whenever PRO-Seq was combined with the dTag system, the infection set included a 6-well plate of confluent HFF infected at MOI of 3 that was used for western blot to determine if the targeted degradation of the viral protein of interest was adequate. On the day before infection, the culture medium was refreshed with 30 mL per T-150 cm$^2$ flask or 2 mL per well of a 6-well plate. On the day of infection, 20 mL of medium in the T-150 cm$^2$ flask was removed and held aside. Virus was inoculated into the 10 mL of medium in the flask. The conditioned medium that was set aside was re-applied after 90 min. For time-course studies, the cells were washed prior to re-applying 30 mL of conditioned medium with balance obtained from uninfected HFF in a T-150 flask. At the time of dTag treatment, 20 mL of medium in the T-150 flask was discarded and 5 mL of the remaining 10mL was removed, mixed with 5 μL of dTag or vehicle control, and then returned to the flask for the indicated time and final concentration of 200 nM dTag. Cereblon (dTag-7)- and VHL-recruiting molecules were kindly provided by Bradner, Winter, and Gray and dissolved in DMSO. For experiments involving treatment with flavopiridol (Flavo; NIH AIDS Reagent Program 9925z, final concentration, 1 μM), 5 mL of medium was removed 1 h before cells were harvested and mixed with 5 μl of

2mM Flavo in DMSO or DMSO alone. The medium was then returned immediately to the flask for a total of 10 mL of culture medium. Phosphonoformic acid (PFA) (Sigma-Aldrich) was added to the medium at a final concentration of 400 μg/mL at the beginning of the 48-h TB40/E WT and 72-h Towne UL87[H] infections. When PFA was added for the last 6 h of the late infection (i.e., PFA treatment for 90–96 hpi), 5 mL of medium was removed, mixed with PFA (final concentration, 400 μg/mL), and returned to the flask. For experiments utilizing 6-well plates, the virus was inoculated in 1 mL of conditioned medium per well and 1 mL of conditioned medium added back after viral adsorption. At the time of dTag treatment, in 1 mL of condition medium was removed, mixed with dTag, and returned to the well.

## Western blot

Sonicated whole-cell extracts in lysis buffer containing phosphatase and protease inhibitors were prepared as described previously [36] and added to gel-loading buffer containing 2% SDS and 100 mM beta-mercaptoethanol. Proteins were fractionated on freshly made SDS-PAGE Tris-glycine gels (8% and 9% gels for UL87[H]F and UL79[H]F, respectively) and transferred for 30 min to Amersham Protran 0.45-μm nitrocellulose membranes (GE Health-care Life Sciences 10600002) using the Thermo Scientific Owl Panther Semidry Electroblotter at 200 mA. HCMV pp28 was detected by mouse monoclonal CMV p28 UL99 antibody (Fitzgerald, 1:500 dilution). HA-tagged proteins (Towne UL87[H], Towne UL87[H]F, TB40/E UL79[H] and TB40/E UL79[H]F) were detected by mouse monoclonal anti-HA.11 epitope tag antibody (BioLegend, 1:1000 dilution). IE1-p72 and IE2-p86 were detected by murine monoclonal antibody MAB810 (EMD Millipore, 1:1,000 dilution). Host actin was detected with polyclonal rabbit anti-actin antibody (Sigma-Aldrich, A2066, 1:4,000 dilution). Primary antibodies were detected with peroxidase AffiniPure F(ab')2 fragment goat anti-mouse IgG (Jackson ImmunoResearch, 115-036-006, 1:40,000 dilution), rabbit anti-mouse IgG (whole molecule)–peroxidase antibody (Sigma-Aldrich, A9044, 1:40000 dilution), or goat anti-rabbit IgG (whole molecule)–peroxidase antibody (Sigma-Aldrich, A0545, 1:40,000 dilution). Blot images were captured and relative differences in intensities of bands quantified using the iBright FL1500 Imaging System (Invitrogen).

## Viral DNA replication assay

HFF were infected in triplicate with Towne WT, Towne UL87[H], Towne UL87[H]F, TB40/E WT or TB40/E UL79[H]F at the indicated MOI. After adsorption of viruses for 90 min, the cells were washed thrice. PFA (Sigma-Aldrich) was added at a final concentration of 400 μg/mL to the medium when indicated. Infected cells were washed with 1x PBS just prior to harvesting in PCR lysis buffer (10 mM Tris-HCl [pH 8.0], 1 mM EDTA, 0.001% Triton X-100, 0.0001% SDS) containing 20 μg/mL proteinase K [37]. The lysate was incubated at 55˚C for 100 min and then heat-inactivated at 95˚C for 20 min. Viral DNA was quantified on the Applied Biosystems 7500 Fast Real-Time PCR System using the standard curve method. Relative quantity of HCMV DNA was normalized to amount of host glyceraldehyde-3-phosphate dehydrogenase (GAPDH) DNA. Primers targeting IE1 exon 4 were described previously [26]. Power SYBR Green PCR Master Mix (Thermo Fisher, 4367659) was used with PCR parameters of 95˚C for 10 min, followed by 40 cycles at 95˚C for 15 s, 60˚C for 60 s.

## PRO-Seq library preparation

Each PRO-Seq library was derived from nuclei isolated from a T-150 cm$^2$ flask. Infected cells were scraped into 10 ml lysis buffer (20 mM HEPES, pH 7.6, 300 mM sucrose, 1% IGEPAL CA-630, 1 mM spermine, 1 mM spermidine, 1 mM EDTA, 1 mM dithiothreitol [DTT], 0.004

U/l SUPERase-In [Ambion AM2696], 0.1% isopropanol-saturated phenylmethylsulfonyl fluoride [PMSF], and cOmplete EDTA-free protease inhibitor cocktail [Roche 11873580001]). Approximately 100,000 moth Sf21 cells (*Spodoptera frugiperda origin)* were spiked into this lysate. Nuclei were prepared, centrifuged through a sucrose cushion, resuspended in storage buffer (20 mM HEPES, pH 7.6, 5 mM magnesium acetate, 150 mM potassium acetate, 5 mM DTT, and 25% glycerol), and stored at -80˚ C, according to the method described in Parida et al [29]. Nuclei were lightly pelleted from storage buffer and resuspended in 40 μL buffer containing 20 mM HEPES (pH 7.8), 100 mM potassium chloride, 5 mM magnesium chloride, 5mM dithiothreitol (DTT), and 0.6 U/μL SUPERase-In (Invitrogen AM2696). Nuclei were then warmed to 37˚C and 20 μL of a 3X nuclear walk-on reaction mix was added. The 3x reaction mix contained 20 mM HEPES (pH 7.8), 5 mM magnesium chloride, 100 mM potassium chloride, 5 mM DTT, 1.5% Sarkosyl, and 60 uM of biotinylated ATP, UTP, GTP, and CTP (Perkin Elmer NEL544, NEL543, NEL545, and NEL542, respectively). After samples were incubated at 37˚C for 10 minutes, 40 μL of 50 mM ethylenediaminetetraacetic acid (EDTA) was added to quench the reactions, and RNA was extracted using 300 μL of Trizol LS (Ambion 10296028), according to the manufacturer's instructions. The protocol described by M. Parida, et al. [29] was used to construct PRO-Seq libraries, which includes steps of RNA base hydrolysis, streptavidin-affinity selection of biotinylated RNA, 3' adapter ligation, 5' adapter ligation, and reverse transcription. We modified the protocol by using RNA adapters having 8-nucleotide UMIs for library construction. The final step of reverse transcription was performed in 23 μL. The amount of cDNA in each library was determined by PCR using the following method. A 2 μL volume of each library was subjected to a series of 4-fold dilutions, and 6 μL of each of 4 dilutions was mixed with 1.5 μM of RP1, 1.5 μM Illumina barcoded index primer RPI-12, and 12.5 μL KAPA HiFi HotStart ReadyMix (KAPA Biosystems KK2601) in a total volume of 25 μL. PCR amplification was performed for 24 cycles using PCR parameters: 98˚C 45 s; 24X 98˚C– 15 s, 60˚C– 30 s, 72˚C– 30 s; 72˚C– 1 min; and 4˚C–hold. S1 Table lists the Illumina barcoded index primers used test and full-scale PCRs. Primers were synthesized and HPLC-purified by Integrated DNA Technologies. PCR products were analyzed on TAE 6% acrylamide gels. The number of PCR cycles needed to produce enough PCR products of the correct size-distribution was applied to the full-scale PCR amplification. This entailed 12 PCR cycles for Towne UL87HF experiment 1, 13 PCR cycles for Towne UL87HF experiment 2, 13 PCR cycles for Towne UL87H PFA experiment, 13 PCR cycles for TB40/E UL79HF experiment 1, 14 PCR cycles for UL79HF experiment 2, and 13 PCR cycles for TB40/E time course experiment. For full-scale PCR, the remaining 21 μL of cDNA library amplification was mixed with 1.5 μM RP1, 1.5 μM of an Illumina barcoded index primer (i.e., RPI 1–12, RPI-18, RPI-19, RPI-22, RPI-25, RPI-27, or RPI-29), and 25 μL KAPA HiFi HotStart ReadyMix in a total volume of 50 μL. PCR products were purified with a Qiagen MinElute kit in 30 μL volume. The Qubit high-sensitivity dsDNA assay was used to determine the purified cDNA library concentration. A 1 μL volume of each cDNA library was removed and diluted to 2.5 ng/μL for analysis of the cDNA size distribution using the Agilent Bioanalyzer 2100. Equimolar quantities of libraries were pooled and size-selected on a Sage Science Blue Pippen instrument using a 2% agarose gel cassette (BDF2010) with Marker V1 internal standards. Library fragments between 145–600 bp in length were selected. After the size-selection, the pooled library was purified on a Qiagen MinElute column, and the DNA concentration was measured using the Qubit. Pooled libraries were diluted to 2.5 ng/μL for analysis on the Agilent Bioanalyzer. Once the pooled library was confirmed to be size selected correctly, the PRO-Seq library was sequenced. Towne UL87HF experiment 1 PRO-Seq library was sequenced on an Illumina HiSeq 4000 with 150 bp paired end reads. The PRO-Seq libraries corresponding to Towne UL87H PFA experiment and TB40/E UL79HF experiments 1 and 2 were sequenced on the Illumina NovaSeq 6000 SP, with 50 bp

paired end reads. The TB40/E time course PRO-Seq library was sequenced on an Illumina Nova-Seq 6000 SP, with 65 bp paired end reads. Sequencing was performed at the University of Iowa Genomics Division except for the experiment 2 Towne UL87[H]F PRO-Seq library that was sequenced at BGI Genomics on an Illumina PE150-Xten with 150 bp paired-end reads.

## ChIP-Seq

HFF infected in T-150 cm$^2$ flasks at MOI of 3 (Towne UL87[H] and TB40/E UL79[H]F) were subjected to formaldehyde crosslinking at 72 h pi. This entailed adding 16% paraformaldehyde (Electron Microscopy Sciences 15710) to a final concentration of 1% for 10 min at room temperature. The crosslinking was stopped by adding Tris (pH 7.6) at final concentration 1.33 M. Cells were washed twice with ice-cold PBS and pelleted at 1200 x g for 5 min at 4˚C. Pellets were mixed with 1 mL ice-cold ChIP buffer (10 mM Tris pH 7.6, 150 mM NaCl, 1% Triton X-100, 1 mM EDTA, 0.25% sodium deoxycholate, 1mM DTT, 0.1% isopropanol-saturated PMSF and cOmplete EDTA-free protease inhibitor cocktail) and sonicated for 25 cycles at 30% amplitude (30 s-on, 30 s-off) by Qsonica Q800R3 Sonicator. After sonication, samples were pelleted at 16000 x g for 15 min at 4˚C and the supernatants were divided equally for use in immunoprecipitations. Approximately 7.5 x 10$^6$ HFF cells were used for each IP. Two separate 20 µL volumes of Pierce Protein A/G Magnetic Beads (Thermo 88802) were used for each ChIP after the beads have been washed and blocked. A 20 µL bead volume was washed with 500 µL ChIP buffer (10 mM Tris pH 7.6, 150 mM NaCl, 1% Triton X-100, 1 mM EDTA, 0.25% sodium deoxycholate, 1mM DTT, 0.1% iso-propanol-saturated PMSF and cOmplete EDTA-free protease inhibitor cocktail) and then blocked in 500 µL ChIP buffer containing 1 mg/ml bovine serum albumin (BSA, RPI A30075) for 1 h or overnight at 4˚C with rotation using a HulaMixer Sample Mixer (Invitrogen, 15920D). The sonicated fixed chromatin sample was first incubated for 2 h 4˚C with rotation with the beads that had blocked for 1 h. The beads were then removed using the DynaMag-2 Magnet (Invitrogen, 12321D). The sonicated chromatin sample was placed in a new tube and incubated with 10 µg anti-HA antibody (Cell Signaling C29F4) or 2.5 µg Pol II antibody (Santa Cruz Biotechnology sc-55492) overnight at 4˚C with rotation. The blocked beads that had been incubated overnight were washed with 500 µL ChIP buffer and, to these, the sonicated chromatin sample containing primary antibody was added and incubated for 2 h at 4˚C with rotation. Samples were then washed 5 times with 500 µL wash buffer (10 mM Tris pH7.6, 150 mM NaCl, 1 mM EDTA, 1% Triton X-100, 0.1% sodium deoxycholate and 0.1% SDS), and two times with 500 µL rinse buffer (10mM Tris pH 7.6, 50 mM NaCl and 1 mM EDTA). Beads were resuspended in 100 µL elution buffer (10 mM Tris pH 7.6, 1 mM EDTA, and 1% SDS) for 10 min at 65˚C. Eluates were treated with 2 µL RNase A (Thermo EN0531) for 30 min at 37˚C, then 2 µL Proteinase K (Thermo EO0491) for 2 h at 65˚C. DNA was purified using the MinElute PCR Purification Kit (Qiagen). The concentration of each sample was determined by the Qubit high-sensitivity dsDNA assay and 10–20 ng was used to generate sequencing libraries using the xGen Prism DNA Library Prep Kit (IDT, 10006202) and xGen UDI Primer Pairs (IDT, 10005975). Library DNA size distribution was analyzed by the Agilent Bioanalyzer 2100. Libraries were pooled in equimolar concentration and size-selected on a Sage Science Blue Pippen instrument using a 2% agarose gel cassette (BDF2010) with Marker V1 internal standards. Fragments between 200–600 bp in length were selected for sequencing that was performed by the Iowa Institute of Human Genetics Genomics Division on an Illumina NovaSeq 6000 SP, with 50 bp paired end reads.

## Bioinformatics

All PRO-Seq datasets were processed (adapters trimmed, aligned to concatenated genome containing Towne FJ616285.1, TB40/E KF297339.1, hg38, and JQCY02.1, deduplicated,

normalized, and converted to bigwigs) as described [26]. Note that libraries were prepared with 8N UMI RNA adapters, so a length parameter of 34 was used for trimming and–n parameter of 8 for dedup. These spike-in controlled datasets were normalized as described in Ball et. al. [38] and can be found in **S2 Table**. Strand specific five prime tracks were generated using bedtools genome coverage [39] and bedGraphToBigWig [40]. Demultiplexed ChIP-Seq sequencing data (fastq files) were first trimmed of adapters using trimgalore—paired—small_rna—dont_gzip—quality 0—length 34 (trim_galore 0.4.4 https://github.com/FelixKrueger/TrimGalore) to obtain a minimum insert size of 18 bp. Next, Bowtie was used to map trimmed reads to the same concatenated genomes as above containing FJ616285.1/KF297339.1 and hg38 using the following parameters:—trim5 8—trim3 8—minins 34—maxins 600—fr—best—allow-contain—sam—fullref—threads 80. Bed files were generated using dedup (https://github.com/P-TEFb/dedup) with a length parameter (-n) of 8 and the flip strands parameter (-f). Bed files were converted to bigwigs using bedSort, bedtools genome coverage and bedGraphToBigWig then visualized on track hubs in the UCSC genome browser with the PRO-Seq data. The ChIP-seq datasets can be found in **S2 Table**.

## Heatmaps of size-sorted ChIP-Seq DNA fragments

Heatmaps were generated by counting the total number of fragments for a specific fragment size between 18 and 400 bp for each position within a genomic interval. The entire HMCV genome was divided into 21,000 bp genomic intervals with 1000 bp overlap between consecutive intervals. These intervals were then used to make fragment heatmaps for presenting a broader view of transcription. A single pixel was maintained for each genomic position along the horizontal axis and 6 pixels along the vertical axis for each fragment size. Major tick marks were added every 500 bp and minor tick marks were added every 100 bp. Along the vertical axis fragment sizes were sorted from short to long order. Major and minor tick marks were added every 50 and 10 bp apart. Additionally, 5 pixels for each tick mark of 30 bp size was maintained on the horizontal and vertical axis. The values at each horizontal position of the fragment sizes were used to assign intensities using the gray colors function in R. A linear relationship between relative read value and intensity was utilized. Black was set at max read value. Finally, gamma correction of 0.5 was applied to enhance the perception of dark and light features on heatmaps. Similarly, fragment heatmaps were generated for genomic intervals centered on specific MAXTSS with a few differences. A single pixel was maintained for each genomic position along the horizontal axis and 4 pixels along the vertical axis for each fragment size. 100 bp major and 50 bp minor tick marks were added across the horizontal axis. Furthermore, two major tick marks were added to indicate the ±1 bp region. Finally, 3 pixels and 20 bp size was maintained for each tick mark.

## Measurement of UL87 LTF ChIP-Seq peak size

Paired-end sequencing of ChIP-Seq fragments enabled the determination of the midpoint base position (center) of each fragment. The fragment centers (Frag Centers) were mapped to the Towne FJ616285.1 genome. The size of a UL87 LTF ChIP-Seq peak was measured by counting the number of fragment centers, for fragments of 100–200 bp in length, that are located within a 200 bp window centered on base position -25 relative to the MAXTSS of the LTF-responsive promoter TSR. Fragments of 100–200 bp in length accounted for 29.3% of the all the deduplicated fragments. The center for an even-sized fragment was assigned 2 bases and each base carried a value of 0.5. An odd-sized fragment had a 1-base center that carried a value of 1. Bedtools v2.26.0 was used to count the total number of overlapping fragments with centers in the 200 bp window. To view results in the UCSC Genome Browser, the fragment

center data was converted from bed format into bedGraph and subsequently into bigwig format. Bigwig files were uploaded to GitHub Towne UL87-LTF repository for visualization.

## Quantitative analysis of viral transcription

Viral TSRs across the HCMV Towne genome were found as described in M. Li, et. al. [26] using the PRO-Seq dataset for HCMV UL87[H]F CTRL (No dTag) Flavo Experiment 1. Viral TSRs with more than 200 total 5' end reads within the 20 bp regions were used to quantify 5' end reads in these 20-bp regions from other datasets. For the HCMV TB40/E UL79[H]F analyses, the same analytical approach was applied with exception that experiment Experiment 1 No dTag PRO-Seq-Flavo dataset was used to generate TSRs on the HCMV TB40/E genome. PFA Sensitivity Indexes (PSIs) were calculated by quantifying 5' end reads in 72 hpi Flavo and 72 hpi Flavo PFA datasets for the Towne UL87[H]F and TB40/E UL79[H]F TSRs and making a column for 72 hpi Flavo/72 hpi Flavo PFA. Division by zero errors were avoided by changing 72 hpi Flavo PFA TSRs with zero reads to one. To determine the ratio of the number of 5' ends at the MAXTSS with no dTag over 6-h dTag, viral TSRs were identified using a 20 bp TSR with at least 20 reads that were 600 bp or less with an average read length of at least 30 bp using the tsrFinderM1 program (https://github.com/P-TEFb/tsrFinderM1). A MAXTSS was generated for each TSR by choosing the TSS with most 5' end counts within a specified TSR window. A list of non-overlapping TSRs for each HCMV no dTag dataset were retained for this analysis. The total number of MAXTSS reads was generated using bedtools v2.26 intersect program with same strandedness option, multiplied with their sample associated spike-in correction factor, and lastly the no dTAG counts were divided over 6-h dTag counts. 6-h dTag values of zero were replaced with 1 to avoid ratios of infinity.

## Comparison of HCMV Towne and TB40/E TSRs

The one-to-one comparison of the effect of LTF depletion on TSRs of HCMV Towne BAC and TB40/E BAC viruses was performed on each viral TSR measuring >200 reads in the CTRL-treated infections and conserved between the two viruses. **S1 Data** lists each TSR >200 reads for CTRL-treated HCMV Towne BAC (72 and 96 hpi) and TB40/E BAC (72 hpi) viruses, according to TSR strength (number of reads) and TSR base position in Towne FJ616285 and TB40/E KF297339 genomes. The genome position for each Towne BAC virus TSR is listed alongside the genome position for the corresponding TB40/E BAC virus and vice versa. The dataset also lists the TSRs present in Towne BAC virus but not in TB40/E BAC virus and vice versa. Caution should be used in comparing viruses with respect the strength of individual TSRs because Towne and TB40/E infections, sample processing, library construction, and sequencing were not performed in parallel.

## Quantitative analysis of host transcription

We applied truQuant differential expression analysis to quantify change transcription at active host genes [26]. Reads from Towne UL87[H]F Flavo No dTag Experiment 2 (72 hpi) and TB40/E UL79[H]F Flavo No dTag Experiment 1 (72 hpi) PRO-Seq datasets were combined and processed by a new version of truQuant (https://github.com/meierjl/truQuant). Other PRO-Seq datasets were also supplied to truQuant to be quantified using the annotation generated from the combined datasets. Columns containing data from other datasets were then normalized. Genes containing less than twenty 5' end reads in the pause region in the combined dataset were discarded for further analysis. The new version of truQuant includes expanded blacklisting (LSU and SSU RNAs are now blacklisted), increased TSR size, and improved pause region centering. Briefly, this is completed by mapping TSRs to search regions as done previously,

except TSRs are now 150 bp instead of 20 bp. truQuant now defines a 150 bp pause region centered around the weighted center of the TSR. Additionally, truQuant annotation and quantitation are combined so one command can be used. Finally, the program is now optimized and multi-processed to quantify multiple datasets quickly.

## Promoter motif analysis

The MEME motif discovery tool was applied to viral promoter regions positioned -50 to -20 bp upstream from the MAXTSS. The MAXTSS position was verified by 5' PRO-Cap plus Flavo results from the dataset for HCMV Towne infection of HFF at 96 hpi [29]. Promoter regions of designated groups of viral promoters were analyzed using the MEME Suite 5.3.2 classic mode (multiple occurrences of a motif in a sequence, motif length is from 4 to 50 and only searching the given strand), as well as the Differential Enrichment mode.

## Supporting information

**S1 Fig. Lytic cycle protein expression and DNA replication of HCMV Towne UL87-tagged viruses.** HCMV Towne UL87H and UL87HF at equivalent infectious units were applied to HFF. Whole cell lysates were analyzed at 24, 48, 72, and 96 hpi by (**A**) western blot (MOI of 3) with antibodies against the indicated viral and host proteins; and (**B**) qPCR (MOI of 0.5) to quantify HCMV genomes after normalization to host GAPDH DNA. Infections were also carried out for 72 h in presence of PFA to prevent viral DNA replication. Relative change in viral DNA level was computed using standard curve method; depicted as mean ±SD for 3 separate infections per group. (**C**) HCMV Towne WT and UL87HF DNA replication was analyzed at 12 and 96 hpi, according to method described in panel B. (**D**) The effect of 6-h dTag1 treatment on HCMV UL87HF DNA replication was measured by qPCR, using the method described above, and the results displayed relative to CTRL treatment (- dTag). The dTag1 (200 nM) was applied for the final 6 h of the 72 or 96 h infection. (**E**) dTag 1 (200 nM) was applied for the final 2 h and 6 h of the 96-h infection with HCMV UL87HF (MOI of 3). Vehicle (CTRL) was applied for the final 6 h. Western blot was performed with antibodies against UL87 HF, IE2-86, and late proteins IE2-40, IE2-60, and pp28. MK, mock infection. This is the same infection analyzed in **Fig 7F**.
(TIF)

**S2 Fig. Construction and phenotype validation of HCMV UL79-tagged viruses, UL79H and UL79HF.** (**A**) Both viral constructs have an HA epitope (H) fused in frame to the amino end of the UL79 ORF. UL79HF has FKBP12 (F) fused in-frame to the carboxy end of the UL79 ORF. (**B**) Electrophoretic pattern of Bam HI fragments from genomes of the viral constructs and the parent WT virus. Astericks mark the restriction fragments containing the UL79 ORF. (**C**) Time course of tagged UL79 protein expression by UL79H and UL79HF in HFF at MOI of 3. The same western blot was re-probed for viral IE1, IE2, late protein pp28, and host actin. MK, mock infection. (**D**) Relative amount of UL79HF vs WT DNA at 24, 48, 72, and 96 h pi in HFF at MOI of 0.5. Results represent 3 biological replicates normalized to amount of host GAPDH DNA. PFA was present throughout the 72-h infection in a parallel set of infections. (**E**) The same protein extracts analyzed in **Fig 1C** were analyzed by western blot with antibodies against UL79HF, IE2-86, and late proteins IE2-40, IE2-60, and pp28.
(TIF)

**S3 Fig. Effect of UL79HF depletion on viral transcription of the MIE gene region.** HFF infected with HCMV TB40/E UL79HF were treated with vehicle control (CTRL) or dTag-2 (dTag) for 6h at 66–72 hpi and Pol II nascent transcripts were then quantified by Pro-Seq

minus and plus Flavo methods. Genome browser of reads aligned to the HCMV TB40/E KF297339 genome. Scale set at 3000 or 5000 reads to allow comparison of majority of viral TSRs. Track arrows point in direction of transcription. Asterisks mark TSRs for IE2-40 (olive green) and EP1 (pink).
(TIF)

**S4 Fig. Concordance in LTF depletion results at 72 vs. 96 hpi and for MAXTSS vs. TSR strength. (A)** Scatterplot of effect of UL87$^H$F LTF depletion (6-h dTag treatment) vs. TSR strength at 72 hpi. Number of nascent RNA reads at each HCMV TSR (TSR strength) were quantified by PRO-Seq-Flavo method. Viral TSRs of >200 reads in CTRL group (open circles) were plotted against the change in TSR strength for dTag vs. CTRL treatment (dTag/CTRL). LTF-activated TSRs (orange dots) represent TSRs decreasing in strength by more than 50% because of dTag treatment. **(B)** Comparison of effects of depleting UL87$^H$F LTF with dTag /CTRL treatment at 72 vs. 96 hpi on 461 viral TSRs (each TSR>200 CTRL reads), as measured by PRO-Seq-Flavo method. 266 TSRs at 72 hpi and 279 TSRs at 96 hpi decrease >50% because of UL87$^H$F LTF depletion. Viral TSRs that decrease in strength by at least 50% at both time points (N = 263) are located in the gold shaded box with dashed borders and account for 57% of all viral TSRs. **(C)** Comparison of effects of depleting UL79$^H$F vs. UL87$^H$F LTF on viral TSRs (6-h dTag /CTRL) at 72 hpi. Viral TSRs conserved between Towne and TB40/E viruses (each TSR>200 CTRL reads) that decrease in strength by >50% (N = 188) are located in the gold shaded box with dashed borders **(D)** Effect of UL79$^H$F or UL87$^H$F depletion on viral MAXTSS and TSR strength for dTag vs. CTRL treatment (6-h dTag/CTRL). The MAXTSS for each viral TSR of >200 reads in CTRL group was included in the analysis.
(TIF)

**S5 Fig. Effect of viral DNA replication blockade on LTF-driven viral TATT and TATAT promoters.** (**A and C**) Scatterplot of PSI vs. effect of UL87$^H$F LTF depletion on TATT (A) and TATAT (C) promoters having TSR strength in top and mid tertiles. A set of 72-h Towne UL87$^H$F infections were treated throughout with or without PFA. Effect of UL87$^H$F LTF depletion was determined from infections described in Fig 3, in which dTag vs. CTRL was added for the last 6 h of infection. Spike-in normalized PRO-Seq-Flavo method was used to quantify change in viral TSRs. (**B**) Scatterplot of viral TSR strength vs. effect of UL87$^H$F LTF depletion on TATAT promoters having TSR strength in top and mid tertiles.
(TIF)

**S6 Fig. Effect of viral DNA replication blockade on LTF-driven viral TGTT and YRYT promoters.** (**A and B**) Scatterplot of PSI vs. effect of UL79$^H$F (A) and UL87$^H$F (B) LTF depletion on TGTT and YRYT promoters having TSR strength in top and mid tertiles. Spike-in normalized PRO-Seq-Flavo method was used to quantify change in viral TSRs, as described in **Fig 6 and S5 Fig**.
(TIF)

**S1 Table. Nucleic acid reagents.**
(PDF)

**S2 Table. PRO-Seq and ChIP-Seq datasets and computations.**
(PDF)

**S1 Data. Corresponding genome positions of Towne and TB40/E promoters and transcription initiation.**
(XLSX)

## Acknowledgments

We thank Jay Bradner, George Winter, and Nathaneal Gray for kindly sharing reagents. We are grateful for assistance and equipment provided by members of the Genomics Division in the Iowa Institute of Human Genetics.

## Author Contributions

**Conceptualization:** Ming Li, Qiaolin Hu, David H. Price, Jeffery L. Meier.

**Data curation:** Ming Li, Qiaolin Hu, Geoffrey Collins, Mrutyunjaya Parida, Christopher B. Ball, David H. Price, Jeffery L. Meier.

**Formal analysis:** Ming Li, Qiaolin Hu, David H. Price, Jeffery L. Meier.

**Funding acquisition:** David H. Price, Jeffery L. Meier.

**Investigation:** Ming Li, Qiaolin Hu.

**Methodology:** Ming Li, Qiaolin Hu, David H. Price, Jeffery L. Meier.

**Project administration:** Jeffery L. Meier.

**Resources:** David H. Price, Jeffery L. Meier.

**Software:** Geoffrey Collins, Mrutyunjaya Parida, Christopher B. Ball, David H. Price.

**Supervision:** Jeffery L. Meier.

**Validation:** David H. Price, Jeffery L. Meier.

**Visualization:** Ming Li, Qiaolin Hu, David H. Price, Jeffery L. Meier.

**Writing – original draft:** Ming Li, Qiaolin Hu, Jeffery L. Meier.

**Writing – review & editing:** Ming Li, Qiaolin Hu, Geoffrey Collins, Mrutyunjaya Parida, Christopher B. Ball, David H. Price, Jeffery L. Meier.

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
