## [Decision Letter · Decision Letter 0]

28 Apr 2021

Dear Dr. Meier,

Thank you very much for submitting your manuscript "Cytomegalovirus Late Transcription Factor Target Sequence Diversity Orchestrates Viral Early to Late Transcription" for consideration at PLOS Pathogens. As with all papers reviewed by the journal, your manuscript was reviewed by members of the editorial board and by several independent reviewers. The reviewers appreciated the attention to an important topic and concluded that the study was well controlled and conducted to a high technical standard. Based on the reviews, we are likely to accept this manuscript for publication, providing that you modify the manuscript according to the review recommendations.

Based on my evaluation of the reviews, I do not think that additional wet lab experiments are needed. Rather, the primary critiques center on clarifying some of your conclusions and inclusion of some additional analyses of your sequencing data. In particular, please address the following points brought up by reviewers 1 & 3:

-Describe any differences in TSR usage between the Towne and TB40/E strains, as these may inform future comparative studies between these strains.

-Provide a figure in which you directly compare the number of LTF ChIP-Seq reads with TSR strength across the viral genome in order to more convincingly address the point that individual promoter-level LTF-driven transcription correlates with the intensity of LTF ChIP-Seq peak reads at the promoter.

-Clarify why depletion of UL87 and UL79 was performed at two different time points (96h versus 72h).

-Per Reviewer 1 point 1, please either temper your conclusion that sequence diversity in the octanucleotide block is the sole regulator of promoter strength or provide additional evidence supporting this claim.

-Describe in more detail how you performed the calculation of promoter activity on a given genome (lines 486-491).

-Please modify figures 6B and 8 to make the scales the same for easier cross comparison.

Sincerely,

Britt A Glaunsinger

Guest Editor

PLOS Pathogens

Klaus Früh

Section Editor

PLOS Pathogens

Kasturi Haldar

Editor-in-Chief

PLOS Pathogens

orcid.org/0000-0001-5065-158X

Michael Malim

Editor-in-Chief

PLOS Pathogens

orcid.org/0000-0002-7699-2064

Reviewer Comments (if any, and for reference):

Reviewer's Responses to Questions

**Part I - Summary**

Reviewer #1: The manuscript by Li et al., used the dTag system to deplete UL87 and UL79, two HCMV proteins, to study sequence variations in promoter motifs responsible for directing transcription of late genes. UL79 and UL87 are part of a six-member viral protein complex referred to as late transcription factors (LTFs). Both proteins are indispensable for transcription of late genes. UL87 is a DNA binding protein that recognizes a distinct TATT element present in late promoters. UL79 is likely to promote RNAPII-mediated transcription elongation of viral genes in late infection. The impact of LTF-depletion was assessed by Pro-Seq-Flavo to determine the amount of RNAPII nascent RNA and the activity of viral transcription start regions (TSRs). Using this approach, the manuscript identified LTF-dependent and -independent promoters. Depletion of UL79 or UL87 abolished transcription of ~60% viral genes but had no impact on transcription of host genes during late infection. Studying sequence variations in LTF-responsive core promoters re-affirmed the importance of the TATTWAA motif that was previously described in KSHV. In addition, the manuscript identified a significant proportion of LTF-responsive promoters with non-TATT motifs. Further fine tuning of these motifs and their correlation with top, mid and bottom TSR activity is described. ChIP-Seq analysis revealed that UL87 and UL79 associate with TATT and non-TATT motifs present in LTF-responsive viral promoters. In summary, the manuscript is well written and uses advanced approaches to identify novel sequence motifs that regulate transcription of viral genes by LTFs. However, as noted below, some conclusions are not fully supported by the data and should be qualified by additional experiments.

Reviewer #2: This is a very carefully conducted study to characterize the cis and trans acting elements that give rise to the diverse pattern of late transcription in HCMV. Many HCMV late promoters are subject to regulation by two late transcription factors (LTF), UL790 and UL87. The authors use a cleaver method to deplete these factors late after infection of HFFs with lab and clinical strains of HCMV. This involves tagging the factors with a moiety that promotes the conditional degradation of the factors by the proteasome. This work was carefully controlled. They follow the utilization of the HCMV promoters as a function of depletion of LTFs by performing pro-seq-flavo to examine the extent of polII binding to TSSs. They also characterized binding of UL79 and UL87 to the viral genome by ChIP-seq. This enabled the characterization of the cis-acing sites for these factors with respect to sequence and strength of action. The main take home of this paper is the diversity of different late promoter types.

This study is carefully conducted and yields important information. There are no issues with the conduct of the experiments or their interpretation. However, it is mostly a characterizational study. The text is very dense with minute details. It is often repetitive between the sections as well.

Reviewer #3: This manuscript provides several significant advances to our understanding of how HCMV regulates transcription from the viral genome, in particular for viral late genes. The authors use cutting edge techniques in combination to provide unprecedented insight into the specific roles of LTFs, as well as the sequence motifs they recognize to control promoter activity. Overall the studies are well controlled, and the data match the conclusions. However some issues to be addressed are noted below.

**Part II – Major Issues: Key Experiments Required for Acceptance**

Reviewer #1: 1- The authors conclude that diversity in LTF target sequences shapes the LTF-transformative program that drives the viral early-late transcription switch. This conclusion is not novel and was reported multiple times in prior publications. Furthermore, the manuscript does not present strong evidence that sequence diversity in the octanucleotide block is the sole regulator of promoter strength. For example, in figure 5B, top, mid and bottom motifs share the TATTT/A sequence. This is also true for figure 7B (TATATA). No experiments were performed to demonstrate the importance of the additional flanking nucleotides. Furthermore, the TATTTTA sequence is present in promoters that markedly vary in their activities suggesting that sequence diversity is not sufficient to regulate promoter strength.

2- The point that individual promoter-level LTF-driven transcription correlates with the intensity of LTF ChIP-Seq peak reads at the promoter, is not properly addressed. Perhaps the author ought to present this point in a stronger manner by providing a table (rather than the link) that directly compares the number of LTF ChIP-Seq reads with TSR strength across the viral genome.

3- Why was depletion of UL87 and UL79 performed at two different time points, 96h versus 72h, respectively?

Reviewer #2: none

Reviewer #3: 1) The authors do not compare TSR usage between the Towne and TB40/E strains. While likely highly similar, this analysis should be included and discuss similarities as well as any differences. This could inform future studies to define mechanistic differences between viral strains.

2) The claim that differential expression of UL1 by the Towne and TB40/E strains is the result of strain-specific nucleotide differences should be further addressed. While there is a correlation with a strong hypothesis, additional evidence (e.g. UL1 promoter reporter plasmids +/- UL79, UL87) would strengthen this claim. Presumably the CHIP-Seq experiments found reduced LTF binding on the UL1 promoter with less activity. While additional experiments would better support the claim, at a minimum this data should be shown.

3) The authors state in the discussion that LTF binding sites identified by CHIP-Seq correlate with LTF-driven transcription. This is an important point, however this analysis is not provided. A genome wide correlation analysis should be included to support this claim.

4) Were UL79 or UL87 associated with host promoters in the CHIP-Seq data? If not, including this point would further strengthen the claim that they do not affect host transcription. If they are associated with host promoters, it would be helpful to discuss what that might mean. In either case, this analysis should be performed, and the results noted in the manuscript.

5) Figure 4 legend states that 4A compares the effect of UL87 and UL79 depletion at 72 hpi. The figure 4 legend also states the same datasets were used as in Figure 3C. The legend for Figure 3C does not note the time after infection for the UL79 and UL867 datasets. However the legend for Figure 3A and 3B states that the data in 3A were from 72 hpi (TB40/E UL79) and data in 3B were from 96 hpi (Towne UL87). The authors should clarify what time points were used for each dataset in each figure.

6) The above point also raises the question of why Figure 3 uses the 72 hour time point for TB40/E-UL79, while 96 hpi was used for the Towne UL87 infection. Data from the same time point after infection for the two viruses should be used in the analysis in Figure 3, especially since the Figure 4 legend states that this data is available.

7) It would helpful in the discussion for the authors to share their thoughts on why host promoter activity was not altered by UL87 or UL79. Why not? Are there no host promoters that contain UL87 or UL79 responsive motifs? If there are, why don’t they respond? I’m sure others would also like to hear the author’s thoughts on this.

8) Lines 486 to 491 provide a calculation of promoter activity on a given genome. The math used to reach these numbers is unclear as described. This section should be rewritten to note the source of the values provided, and better describe the calculations behind these conclusions.

**Part III – Minor Issues: Editorial and Data Presentation Modifications**

Reviewer #1: 1- Figure 1 illustrates the efficiency of dTag depletion using Western blotting. The figure lacks any information on the effect of depletion on the expression level of any late proteins.

2- Figure 3A and 3B, consider changing the figure legend from (Activated) to LTF-responsive. Why are some LTF-responsive genes still active in the absence of UL79 and UL87? The figure needs a cutoff line that denotes the 0.5 value at the x-axis.

3- Figure 5A reveals the presence of 4 TATT genes that are not responsive to LTF depletion. What is different about these four genes?

4- Figure 6B and 8, different scales are used which complicate interpretation of the data.

5- No discussion was provided for the bidirectional promoters in the discussion section.

6- Consider moving lines 187 - 194 to the discussion section.

Reviewer #2: none

Reviewer #3: (No Response)

PLOS authors have the option to publish the peer review history of their article (what does this mean?). If published, this will include your full peer review and any attached files.

Reviewer #1: No

Reviewer #2: No

Reviewer #3: No

Figure Files:

Data Requirements:

Reproducibility:

References:

---

## [Editor Report · Decision Letter 1]

12 Jul 2021

Dear Dr Meier,

We are pleased to inform you that your manuscript 'Cytomegalovirus Late Transcription Factor Target Sequence Diversity Orchestrates Viral Early to Late Transcription' has been provisionally accepted for publication in PLOS Pathogens.

Best regards,

Britt A Glaunsinger

Guest Editor

PLOS Pathogens

Klaus Früh

Section Editor

PLOS Pathogens

Kasturi Haldar

Editor-in-Chief

PLOS Pathogens

orcid.org/0000-0001-5065-158X

Michael Malim

Editor-in-Chief

PLOS Pathogens

orcid.org/0000-0002-7699-2064
---

## [Editor Report · Acceptance letter]

26 Jul 2021

Dear Dr Meier,

We are delighted to inform you that your manuscript, "Cytomegalovirus Late Transcription Factor Target Sequence Diversity Orchestrates Viral Early to Late Transcription," has been formally accepted for publication in PLOS Pathogens.

Best regards,

Kasturi Haldar

Editor-in-Chief

PLOS Pathogens

orcid.org/0000-0001-5065-158X

Michael Malim

Editor-in-Chief

PLOS Pathogens

orcid.org/0000-0002-7699-2064